# Investigating cone photoreceptor development using patient-derived NRL null retinal organoids

Alyssa Kallman[1,11], Elizabeth E. Capowski [2,11], Jie Wang [3], Aniruddha M. Kaushik[4], Alex D. Jansen[2], Kimberly L. Edwards[2], Liben Chen[4], Cynthia A. Berlinicke[3], M. Joseph Phillips[2,5], Eric A. Pierce[6], Jiang Qian[3], Tza-Huei Wang[4,7], David M. Gamm[2,5,8 ✉] & Donald J. Zack [1,3,9,10 ✉]

Photoreceptor loss is a leading cause of blindness, but mechanisms underlying photoreceptor degeneration are not well understood. Treatment strategies would benefit from improved understanding of gene-expression patterns directing photoreceptor development, as many genes are implicated in both development and degeneration. Neural retina leucine zipper (NRL) is critical for rod photoreceptor genesis and degeneration, with *NRL* mutations known to cause enhanced S-cone syndrome and retinitis pigmentosa. While murine Nrl loss has been characterized, studies of human NRL can identify important insights for human retinal development and disease. We utilized iPSC organoid models of retinal development to molecularly define developmental alterations in a human model of NRL loss. Consistent with the function of NRL in rod fate specification, human retinal organoids lacking NRL develop S-opsin dominant photoreceptor populations. We report generation of two distinct S-opsin expressing populations in NRL null retinal organoids and identify *MEF2C* as a candidate regulator of cone development.

[1] Institute of Genetic Medicine, Johns Hopkins University School of Medicine, Baltimore, USA. [2] Waisman Center, University of Wisconsin-Madison, Madison, USA. [3] Department of Ophthalmology, Wilmer Eye Institute, Johns Hopkins University School of Medicine, Baltimore, USA. [4] Department of Mechanical Engineering, Johns Hopkins University, Baltimore, USA. [5] McPherson Eye Research Institute, University of Wisconsin-Madison, Madison, USA. [6] Ocular Genomics Institute, Massachusetts Eye and Ear Infirmary, Boston, USA. [7] Department of Biomedical Engineering, Johns Hopkins University, Baltimore, USA. [8] Department of Ophthalmology and Visual Sciences, University of Wisconsin School of Medicine and Public Health, Madison, USA. [9] The Solomon H. Snyder Department of Neuroscience, Johns Hopkins University School of Medicine, Baltimore, USA. [10] Department of Molecular Biology and Genetics, Johns Hopkins University School of Medicine, Baltimore, USA. [11] These authors contributed equally: Alyssa Kallman, Elizabeth E. Capowski. ✉email: dgamm@wisc.edu; donzack@gmail.com

Normal visual function requires light detection by photoreceptors followed by signal transduction through the neural retina to the brain. Mammalian retinas contain rod and cone photoreceptors, with rods responsible for dim-light and peripheral vision and cones for color, high acuity, and central vision. Rods and cones arise from a common precursor, and photoreceptor cell fate is dictated by key transcription factors[1]. Neural retina leucine zipper (NRL) is required for rod development, and it activates Nuclear Receptor Subfamily 2 Group E Member 3 (NR2E3), which suppresses expression of cone-specific genes, promoting the rod developmental program[2]. Previous murine studies have shown that Nrl loss leads to development of cone dominant retinas; specifically, an increase in S-cones[3].

Like the murine phenotype, loss of NRL in humans can cause enhanced S-cone syndrome, a rare retinal disease characterized by supranormal blue cone function due to an increased proportion of S-cones and night blindness due to the absence of rods[4,5]. However, the range of clinical phenotypes caused by *NRL* mutations is broad, with dominant missense mutations leading to a clinical picture more akin to retinitis pigmentosa[4–6]. Similarly, enhanced S-cone syndrome can result from mutations in genes other than *NRL*, usually *NR2E3*. Using an induced pluripotent stem cell (iPSC) line derived from a patient carrying a homozygous *NRL* mutation, we sought to characterize the developmental and molecular effects of NRL loss in human stem cell-derived retinal organoids. Retinal organoids, which closely mirror in vivo retinal development, provide a human model for studying retinal development and degeneration[7–20]. Organoids closely mimic retinal structure and apical-basal polarity, with an outer layer of photoreceptors capable of ribbon synapse formation and inner layers of retinal ganglion, amacrine, horizontal, and bipolar cells.

In addition to histological characterization of human retinal organoids, transcriptomics, particularly at the single-cell level, can identify and characterize distinct cell populations. Dropseq, a single-cell RNA sequencing (scRNAseq) method utilizing microfluidics and barcoded beads to capture the transcriptomes of single cells, has proven powerful for characterizing mouse retinas and identifying subtypes of mouse bipolar cells[21,22]. More recently, scRNAseq studies with developing and adult retinal tissue have offered insight into in vivo human retinal cell populations[23–25]. While previous studies have utilized scRNAseq to identify cell types of developing retinal organoids, they have not discerned distinct photoreceptor sub-populations[26–28]. Additionally, while transcriptomics have been used to characterize Nrl loss in murine photoreceptors, these analyses were not performed at a single-cell level, thus limiting the potential to identify and characterize sub-populations[29–31].

Here, we present histological and single-cell transcriptomic characterization of human iPSC-derived retinal organoids with and without functional NRL[32]. We show that retinal organoids lacking NRL develop S-opsin+ dominant photoreceptor populations. In contrast to previous studies, we provide evidence that, in the absence of NRL, two distinct populations of S-opsin expressing photoreceptors emerge; one population more representative of typical cones, and the other of rod/cone intermediates. Finally, our analyses identified a putative novel regulator of cone photoreceptor development. This study further defines photoreceptor subpopulations in a human model of NRL loss and provides a platform for characterizing aberrant photoreceptor development.

## Results

### Differentiation of patient and control retinal organoids.
Fibroblasts from a patient with a homozygous null mutation, a frame shift and premature stop (c.223dupC, p.L75Pfs*19) in the *NRL* gene, were reprogrammed to hiPSCs[32]. Three independent, karyotypically normal L75Pfs clones (Supplementary Fig. 1) were compared with WA09 and two wildtype (WT) hiPSC lines (1013 and 1581)[19]. All lines were indistinguishable in their ability to make stage 1 organoids, characterized by a phase bright, neuroepithelial appearance and the presence of ganglion cells and proliferative retinal progenitor cells (Supplementary Fig. 2)[19]. CRX+/RCVN+ photoreceptor precursor production was comparable between early stage 2 WT and L75Pfs organoids, when photoreceptor subtype specification begins (Fig. 1; compare Fig. 1b, c, merge in e, to g, l, q, h, m, r, merges in j, o and t)[19]. However, NRL+ rod photoreceptors were never detected in L75Pfs organoids (compare Fig. 1d to i, n, s). As photoreceptors matured and formed outer segments (the "hair-like" surface projections in Fig. 2c, i), L75Pfs organoids showed a striking S-opsin dominant photoreceptor phenotype (Fig. 2)[19]. Unlike WT organoids, which possess a single layer of ML-opsin+ cones and rare S-opsin+ cones along the outermost aspect of the outer nuclear layer (ONL) (Fig. 2a), L75Pfs organoids contained S-opsin expressing cells throughout the ONL and in greater abundance than ML-opsin expressing cells (Fig. 2b; also compare Fig. 2e, g to k, m). Quantification of ARR3+ cones and NR2E3+ rods as a percent of total nuclei in the ONL (Fig. 2d–h, j–n) revealed a dramatic reduction in rods and an increase in cones in the L75Pfs ONL (Fig. 2o). Interestingly, while WT organoids had rare ARR3-/NR2E3- nuclei in the ONL, ~20% of the L75Pfs ONL nuclei expressed neither marker. Since ARR3 is normally expressed >60 days after cone progenitors are detected, these ARR3-/NR2E3- cells may represent rod progenitor-derived cells that either have not committed to a cone fate or do not yet express ARR3. We quantified the ML- or S-opsin expressing cells as a fraction of the total ARR3+ cells and detected a 38-fold shift in the ML:S-opsin cone ratio, from 19:1 in WT to 1:2 in L75Pfs organoids (Fig. 2p). Additional analyses of rod and cone gene expression by RT-qPCR revealed that rod developmental genes were downregulated in L75Pfs organoids relative to WT organoids, while S-opsin expression was significantly increased in the L75Pfs organoids (Fig. 2q, $p < 0.005$, Mann–Whitney test). Thus, in L75Pfs human retinal organoids, rods appeared shifted toward an S-cone fate, consistent with the *Nrl*−/− mouse phenotype[3,33]. Finally, we examined the inner nuclear layer (INL) of L75Pfs organoids and found it indistinguishable from WT organoids (Supplementary Fig. 3a–n), including comparable production of PKCα+ rod bipolar cells (consistent with the *Nrl*−/− mouse phenotype)[34]. However, in contrast to the *Nrl*−/− mouse, L75Pfs organoids displayed an intact outer limiting membrane (OLM) with no increase in rosette formation compared to WT organoids (Supplementary Fig. 3o–y)[3,35].

### Reintroduction of WT NRL restores rod formation.
To confirm that the observed L75Pfs phenotypes were indeed due to lack of NRL, we introduced functional *NRL* to determine whether NRL expression could rescue the phenotype. We ectopically expressed WT *NRL* in d90 L75Pfs retinal organoids using a lentivirus expression cassette. Of note, d90 corresponds to the onset of NRL protein detection in WT organoids[19]. After 100 additional days in culture, organoids transduced with virus containing either a control *GFP* expression cassette (without *NRL*) or a WT *NRL* expression cassette were examined by immunocytochemistry for NRL, rhodopsin (RHO), and S-opsin expression. Figure 3a–d shows co-expression of GFP in ARR3+ cones of control cassette-treated L75Pfs NRL organoids, which remained NRL-/RHO- and expressed S-opsin throughout the ONL (Fig. 3e–l), similar to untreated L75Pfs organoids. In contrast, WT *NRL* expression

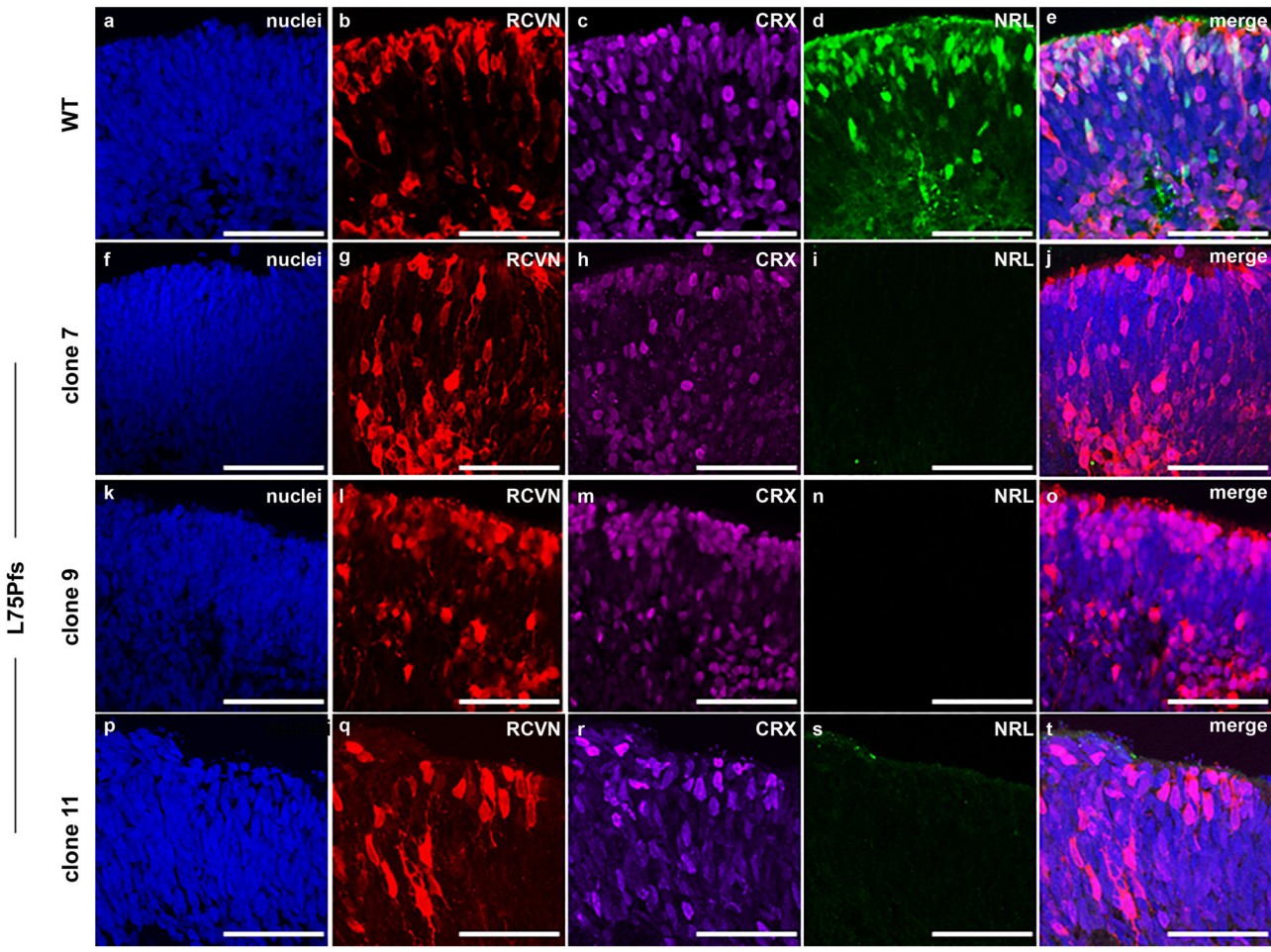

**Fig. 1 Photoreceptors from L75Pfs retinal organoids lack NRL protein expression. a–t** Confocal images of d100 (stage 2) organoids from a WT line (**a–e**) or three individual clonal lines of the L75Pfs mutant (**f–t**) showing photoreceptors immunostained for RCVN (**b**, **g**, **l**, **q**), CRX (**c**, **h**, **m**, **r**), or NRL (**d**, **i**, **n**, **s**). **a**, **f**, **k**, **p**: nuclei (blue); **e**, **j**, **o** merge in **t**: merge). Scale bars = 50 μm.

cassette-treated organoids showed NRL protein in patches of nuclei within the ONL (Fig. 3m–t). Furthermore, all cells with restored NRL expression did not express S-opsin (Fig. 3m–t). Additionally, rare RHO+ cells (Fig. 3q, w), which were never observed in untreated or pgkGFP-transduced (Fig. 3i–l) L75Pfs organoids, were observed and were uniformly negative for S-opsin (Fig. 3t) and ARR3 (Fig. 3x). Of note, the localization of RHO to outer segments in some lenti-pgkNRL transduced cells (Fig. 3w) is reminiscent of RHO immunostaining in WT organoids (Supplementary Fig. 4a–d). Thus, restoring NRL protein expression to L75Pfs photoreceptor precursor cells restricted S-opsin expression and could promote, although at low efficiency, RHO expression.

**scRNAseq to identify and analyze organoid cell types.** After establishing that the observed L75Pfs phenotype was due to NRL loss, we sought to identify and transcriptomically analyze the cell populations in WT and L75Pfs retinal organoids. We performed scRNAseq via the Dropseq platform on WT and L75Pfs organoids differentiated to early (100–103 days) or late (170 days) stage 2[19,21]. At the earlier time, 4 WT and 4 L75Pfs organoids yielded transcriptional profiles of 5294 and 4787 cells, respectively. Cells were clustered by t-distributed stochastic neighbor embedding (tSNE) using the Seurat R package[36]. The even distribution of cells classified either by number of genes expressed or number of unique molecular identifiers (UMIs) throughout

the clusters confirmed that these factors were not driving clustering (Supplementary Fig. 5). Rather, based on known marker genes (Supplementary Table 1, Supplementary Fig. 6), the clusters represent stereotypical retinal populations present in both WT and L75Pfs organoids (Fig. 4a). Both rod and cone photoreceptors were present, with almost all *NR2E3* expressing cells being WT (Fig. 4b). Spearman correlations were performed between WT cells of each population and published fetal and adult retinal scRNAseq datasets (Supplementary Figs. 7 and 8)[23,24]. This analysis revealed d100 organoids yielded amacrine, horizontal, and retinal ganglion cells more similar to fetal retinal populations. Rods most closely resembled adult peripheral rods, while cones and Müller glia more closely resembled adult foveal cells. Differential gene expression tests were performed between WT and L75Pfs cells of each cluster, and genes with significantly different expression and an average natural log fold change greater than 0.5 (~1.6 fold) are summarized in Supplementary Data 1. Of genes enriched in L75Pfs cells of the rod cluster, the presence of the cone transducin, *GNGT2*, indicates that this population is acquiring a cone-like profile.

At d170, 3 WT, and 6 L75Pfs organoids yielded 8920 and 15,447 single-cell transcriptomes, respectively. Cell populations identified by marker gene expression (Supplementary Table 1) showed that mature retinal cells were captured, including bipolar cells and opsin-expressing photoreceptors (Fig. 4c, Supplementary Fig. 9). Loss of retinal ganglion cells was also observed.

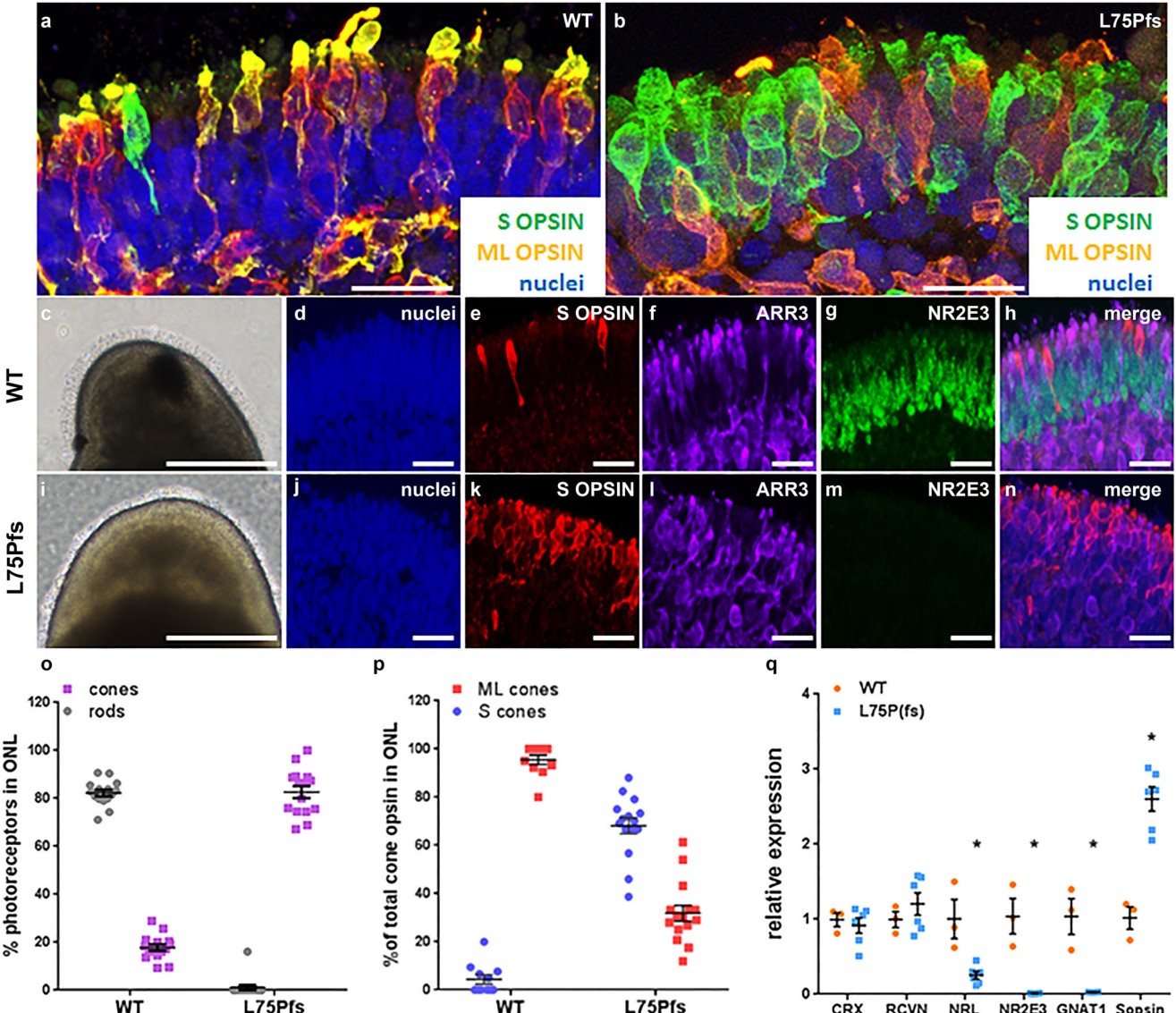

**Fig. 2 L75Pfs retinal organoids display an overabundance of S-opsin expressing cells at the expense of rods compared to WT organoids. a, b** Confocal images from stage 3 organoids (i.e., presence of photoreceptor outer segments) showing a single layer of cones with few S-cones (green) in WT organoids (**a**) versus an abundance of S-cones (green) distributed throughout the ONL in L75Pfs organoids (**b**). ML-cones are shown in orange. Scale bars = 25 μm. **c–n** Photoreceptor characterization of WT and L75Pfs retinal organoids. Bright field (**c, i**) and confocal (**d–h** and **j–n**) images showing S-opsin+/ARR3+ cones (**k, l**) distributed throughout the ONL of L75P(fs) organoids that do not express the rod marker NR2E3 (**m**) (a transcription factor whose expression is controlled by NRL). This finding is in contrast to WT organoids that display ordered expression of cones (**e, f**) along the outermost ONL with a multicellular layer of NR2E3+ rod nuclei (**g**) internal to the cone layer, as well as an overall low number of S-opsin+ (**e**) cones. Scale bars: **c, I** = 250 microns; **d–h** and **j–n** = 25 μm. **o, p** Quantification of photoreceptors in confocal images of stage 3 organoids from 3 WT lines and 3 L75Pfs clones. **o, p** Quantification of photoreceptors in confocal images of stage 3 organoids from 3 WT lines and 3 L75Pfs clones. **o** NR2E3+ rod and ARR3 + cone abundance as a percentage of total nuclei in the ONL: 15 images from 5 organoids per line or clone were counted. **p** ML- and S-cone abundance as a percentage of total ARR3+ cones in the ONL: 11 WT images from 4 organoids per line and 15 L75Pfs images from 5 organoids per clone were counted. **q** RT-qPCR from stage 2–3 organoids showing a reduction in rod transcripts and an increase in S-opsin transcripts in L75Pfs organoids relative to WT organoids. *p* < 0.005, Mann–Whitney test.

Possible explanations for retinal ganglion cell loss include microfluidic bias favoring other cell types or death of retinal ganglion cells due to the lack of vasculature in retinal organoids. Notably, age-dependent retinal ganglion cell loss has been reported in retinal organoids[19]. Again, Spearman correlations were performed between WT cells of each population and published fetal and adult retinal scRNAseq datasets (Supplementary Figs. 10 and 11)[23,24]. Like d100, organoids at d170 yielded amacrine and horizontal cells more similar to fetal cells. Rods and bipolar cells more closely resembled adult peripheral cells, and,

similarly to d100, cones and Müller glia were more highly correlated with adult foveal cells. Two cone opsin-expressing populations were identified, one that expressed both ML-opsin and S-opsin and the other consisting of cells that primarily express S-opsin. Due to the identity of the M- and L-opsin 3' UTRs that were captured via our analysis, we could not distinguish between M- and L-opsin transcripts. Interestingly, 1.5% of WT cone opsin-expressing cells co-expressed both ML- and S-opsin (Fig. 4d). The number of UMIs and genes expressed by these cells suggests they are not doublets (Fig. 4e).

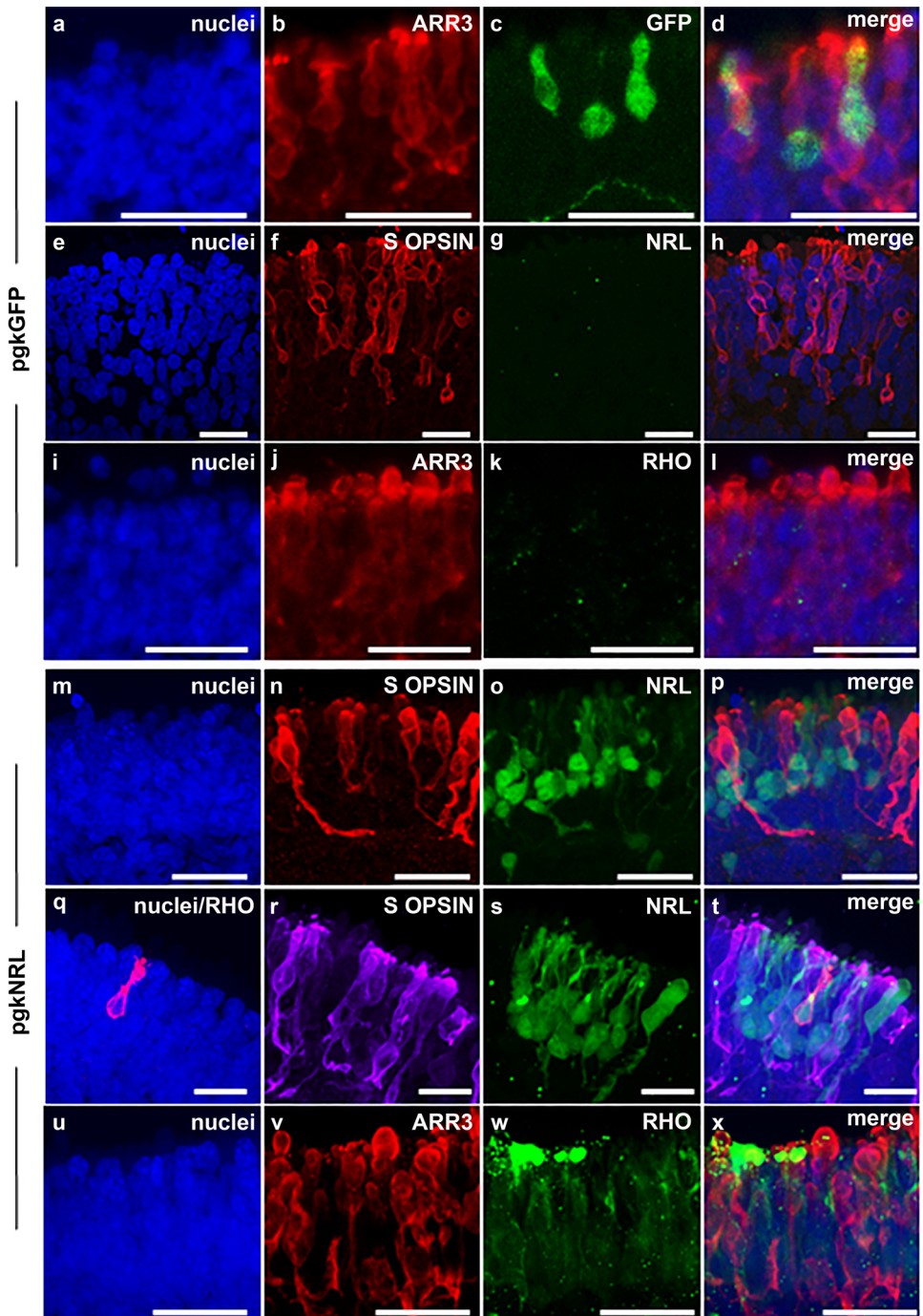

**Fig. 3 RHO expression is found in L75Pfs organoids following viral transduction with an *NRL*-expression cassette. a–h** Confocal images from L75Pfs organoids transduced with a control lentivirus carrying a pgkGFP expression cassette. GFP (**c**) was found in ARR3+ (**b**) cones; NRL (**g**) was not detected, and S-opsin+ cones (**f**) were localized throughout the ONL and no RHO (**k**) expression was detected, as expected in the absence of ectopic WT NRL expression. **m–x** Confocal images from L75Pfs organoids transduced with lentivirus carrying a pgkNRL expression cassette reveal patches of NRL expression within the ONL (**o** and **s**) and no NRL co-expression with S-opsin (**n** and **r**; merges in **p** and **t**). However, ectopically expressed NRL does co-express with RHO (**q**; merge in **t**), a rod marker that was never present in control transduced L75Pfs organoids. In pgkNRL-transduced organoids, RHO (**w**) did not co-localize with the cone marker ARR3 (**v**; merge in **x**). Scale bars = 25 μm.

We performed differential gene expression analysis on these 3 cone opsin-expressing populations (ML-, ML/S-, or S- expressing) to identify novel markers of developing cone subtypes. As expected, there were only minimal differences in gene expression between the cone populations. We identified *MYL4* as a possible ML-cone marker and *CCDC136* and *DCT* as possible S-cone markers (Fig. 4f). *CCDC136* is preferentially expressed in mouse S- and S/M-cones and recently has been shown to be enriched in primate S-cones[37,38]. Interestingly, Peng et al. identified *MYH4*, the heavy chain complement to MYL4, as a transcript distinguishing ML-cones from S-cones[38]. *NUP93, SLC12A6, PDRG1,* and *TRAPP2CL* were significantly enriched in the ML/

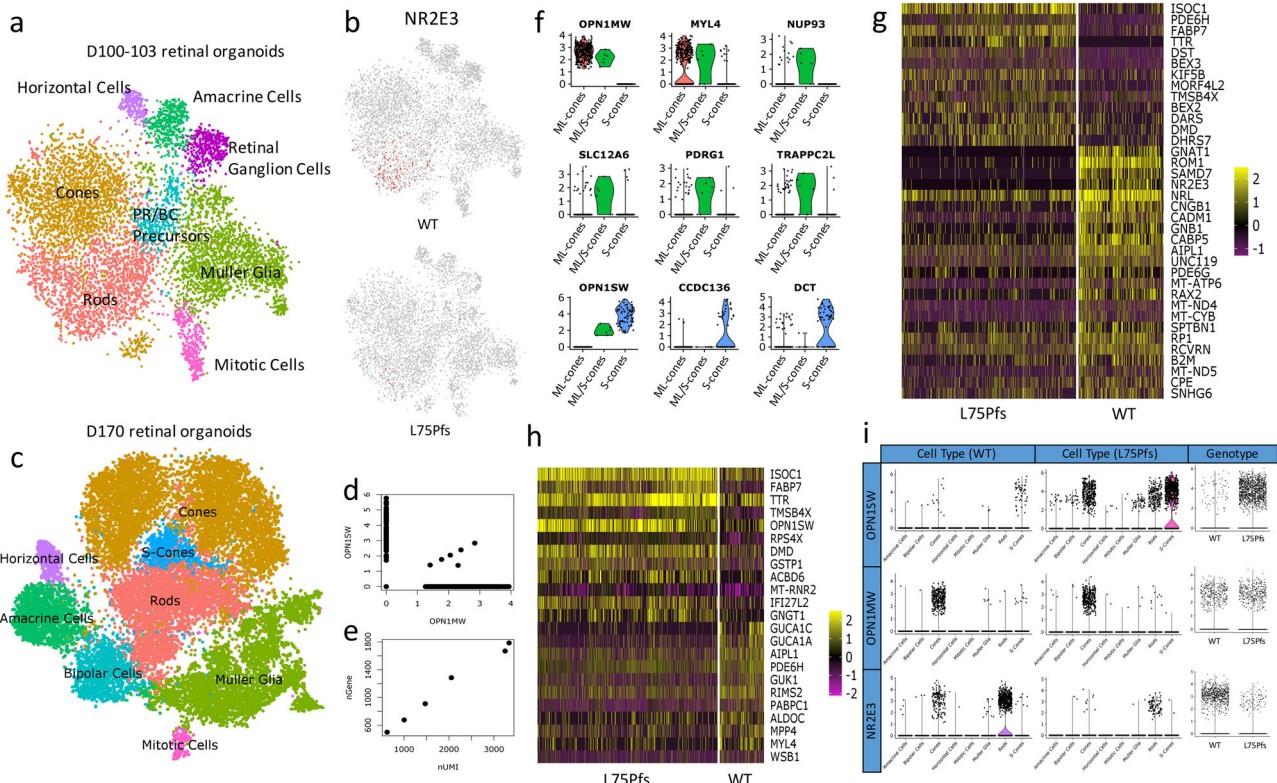

**Fig. 4 Identification and characterization of cell populations present in WT and L75Pfs retinal organoids by scRNAseq. a** tSNE plot showing cell populations present in all d100 organoids. **b** Expression of *NR2E3* in WT (top) compared to L75Pfs (bottom) cells, showing that nearly all expression is in WT cells. **c** tSNE plot showing cell populations present in all d170 organoids. **d** Scatter plot of *OPN1SW* and *OPN1MW* expression at d170 indicating six co-expressing cells. **e** Scatter plot showing the number of UMIs and genes expressed by the six co-expressing cells from D. **f** Violin plots showing specific enrichment of novel cone marker genes across WT ML-cones, ML/S-cones, and S-cones. **g, h** Heatmap of genes differentially expressed in d170 rod (**g**) and S-cone (**h**) clusters between WT and L75Pfs cells. **i** Comparison of expression of *OPN1SW, OPN1MW,* and *NR2E3* by cell population of WT and L75Pfs organoids, and total expression of *OPN1SW, OPN1MW,* and *NR2E3* within all WT and L75Pfs cells.

S-cone population compared to the ML or S- expressing cell populations. Further studies are necessary to determine if this ML/S-co-expressing cone population exists in vivo.

To determine altered transcriptional profiles at d170, within each cell population we identified differentially expressed genes with an average natural log fold change greater than 0.5 (~1.6-fold) between WT and L75Pfs cells (Supplementary Data 2). Within the rod cluster, WT cells had significantly higher expression of many rod-specific genes including *GNAT1, ROM1, SAMD7, NR2E3, CNGB1, GNB1,* and *PDE6G,* while cone-specific phosphodiesterase, *PDE6H* (Fig. 4g), was enriched in L75Pfs cells, suggesting a more cone-like character of these cells. Despite loss of NRL protein, the *NRL* transcript is still detectable in L75Pfs cells, possibly due to the presence of transcripts that have yet to be removed by nonsense mediated decay. Of differentially expressed genes in the S-opsin expressing population (Fig. 4h), L75Pfs S-opsin expressing photoreceptors were enriched for *OPN1SW* and *GNGT1,* a rod-enriched transducin (Fig. 4g). Despite enrichment of *MYL4* in WT compared to L75Pfs S-cones, this gene exhibited substantially higher expression in ML-cones compared to S-cones, supporting its designation as enriched in ML-cones (Fig. 4f). To identify whether NRL loss alters the distribution of photoreceptor subtypes, we compared expression of *OPN1SW, OPN1MW,* and *NR2E3* within each cell population of d170 WT and L75Pfs organoids, as well as total expression of these genes across both genotypes (Fig. 4i). *OPN1SW* was detected in more cells and more clusters in L75Pfs organoids compared to WT, while *OPN1MW* was primarily detected in only the cone cluster of both the WT

and L75Pfs organoids. Since NRL activates *NR2E3* transcription, L75Pfs organoids had (as expected) lower expression of *NR2E3* and fewer *NR2E3* expressing cells compared to WT. Quantification of *OPN1SW, OPN1MW,* and *NR2E3* expression levels and percentage of expressing cells in d170 organoids can be found in Table 1. Although WT and L75Pfs organoids had significantly different relative numbers of cells expressing *OPN1SW* vs *OPN1MW,* on an individual cell basis the *OPN1SW* and *OPN1MW* expressing cells expressed comparable levels of *OPN1SW* and *OPN1MW.* However, for *NR2E3,* both the percentage of expressing cells and the expression level within individual expressing cells was significantly lower in L75Pfs cells. Taken together, this data suggests that NRL loss has a profound effect on rod development, shifting them towards an S-cone identity.

**Trajectory reconstruction of WT photoreceptor development.** After identifying retinal populations, we used the 5144 WT photoreceptors identified from both time points to create a pseudotemporal trajectory of WT photoreceptor development[39]. While the population contained contaminating bipolar cell precursors (44/5144 cells with *VSX1* or *VSX2* expression), this small population is unlikely to impact trajectory construction (we could have removed these cells, but felt that selectively removing small subpopulations of cells was more likely to produce an artifact than leaving them in). To determine the gene set for ordering the trajectory, we performed a semi-supervised differential gene expression test for genes varying by age and assigned cell type

**Table 1 Percent of d170 cells expressing select rod and cone markers and level of expression of these markers showing altered expression of *NR2E3* and *OPN1SW* in L75Pfs organoids.**

| | % Cells expressing | | Average (normalized) expression level within expression group | | Average (normalized) total expression level | |
|---|---|---|---|---|---|---|
| | WT | L75Pfs | WT | L75Pfs | WT | L75Pfs |
| OPN1SW | **0.796** | **8.021** | 3.648 | 3.793 | **0.029** | **0.304** |
| OPN1MW | **3.711** | **2.278** | 2.612 | 2.654 | **0.097** | **0.060** |
| NR2E3 | **7.735** | **0.667** | 3.084 | 2.657 | **0.239** | **0.018** |

Statistically significant differences are shown in bold.

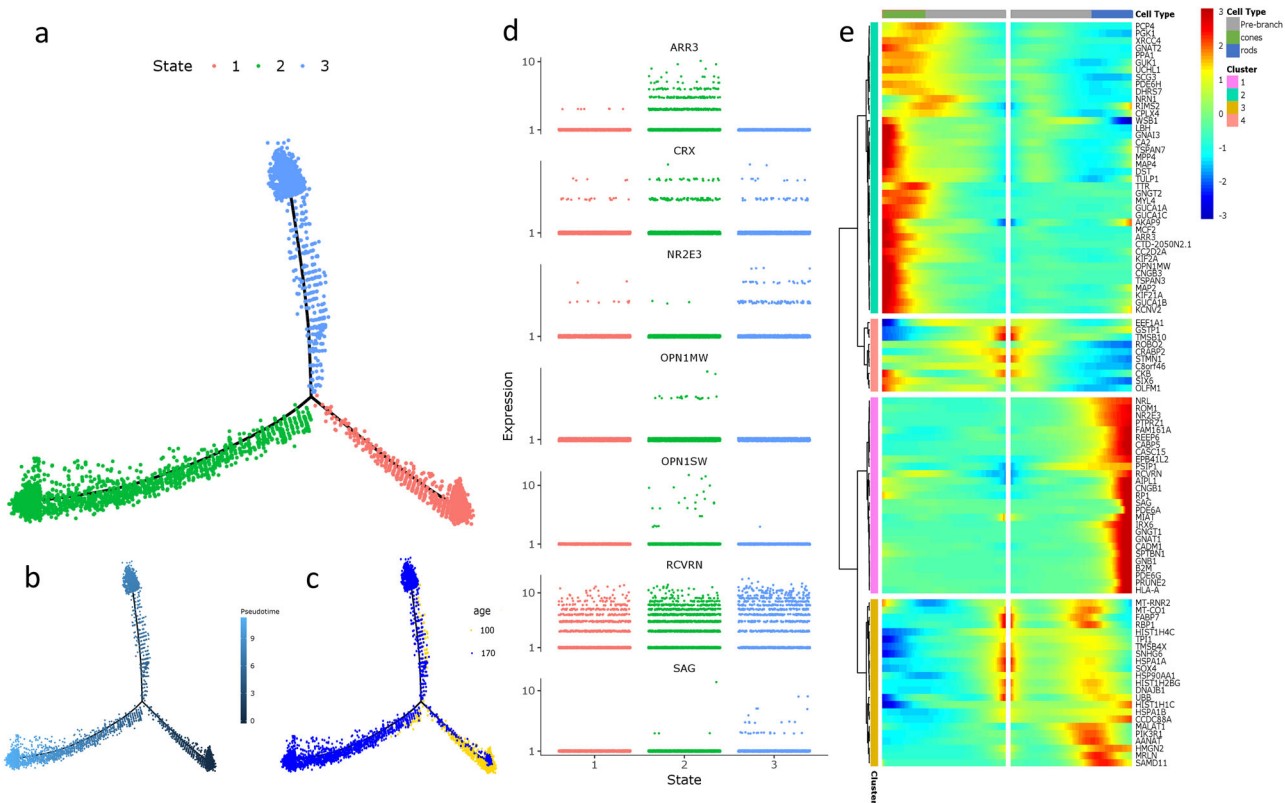

**Fig. 5 Reconstruction of a WT photoreceptor developmental trajectory. a–c** Trajectory of 5144 WT photoreceptors colored by state (**a**), pseudotime (**b**), and age (**c**). **d** Expression of photoreceptor markers used to distinguish the identity of each branch of the trajectory. **e** Heatmap of the top 100 non-ribosomal differentially expressed genes at the node separating rods and cones. Genes are hierarchically clustered into four clusters based on expression pattern. The center of the heatmap is the beginning of pseudotime, with cell maturity moving horizontally to the left (cones) and right (rods).

within the WT photoreceptor dataset. After removing mitochondrial and ribosomal genes, the top 780 genes by *p*-value were used for ordering (Supplementary Data 3). Importantly, neither *VSX1* nor *VSX2* were present in this list, verifying that the contaminating bipolar cells did not affect the trajectory reconstruction. The resulting WT trajectory had one node separating rod and cone photoreceptors, with *OPN1SW* or *OPN1MW* expressing cells in state 2 and *NR2E3/SAG* expressing cells in state 3 (Fig. 5a–d). Five hundred and ninety genes were significantly differentially expressed at this node and the top 100 non-ribosomal genes were used to create a heatmap of genes enriched along the rod versus cone branches (Supplementary Data 4, Fig. 5e). While many of these genes are known as rod- or cone-specific, we identified some novel cone- or rod-enriched genes. In addition to *MPP4* and *CC2D2A*, genes already shown to be enriched in human fetal cones, we identified *GNAI3, CA2, MAP4,*

*MYL4, MCF2, KIF2A,* and *KIF21A* as cone-enriched, and *PTPRZ1, CABP5, IRX6, B2M,* and *PRUNE2* as rod-enriched (Supplementary Fig. 12)[40]. We checked published adult human scRNAseq data and confirmed significant enrichment of *MAP4, MYL4, MCF2,* and *KIF2A* in cones, and *CABP5* and *IRX6* in rods (*p* < 0.05, one-sided *T*-test)[24]. We performed similar trajectory analyses using adult photoreceptor data to compare the organoid trajectory to in vivo photoreceptors (Supplementary Fig. 13)[24]. The resulting trajectory separated rods and cones, and the top 100 non-ribosomal differentially expressed genes between the state 1 rods and state 4 cones were used to create a heatmap for comparison to organoid development. Thirty genes were differentially expressed in both datasets, with all but *MALAT1, TMSB10,* and *EEF1A1* exhibiting the same enrichment pattern. The expression patterns of known markers confirm that our trajectory accurately recapitulates photoreceptor development

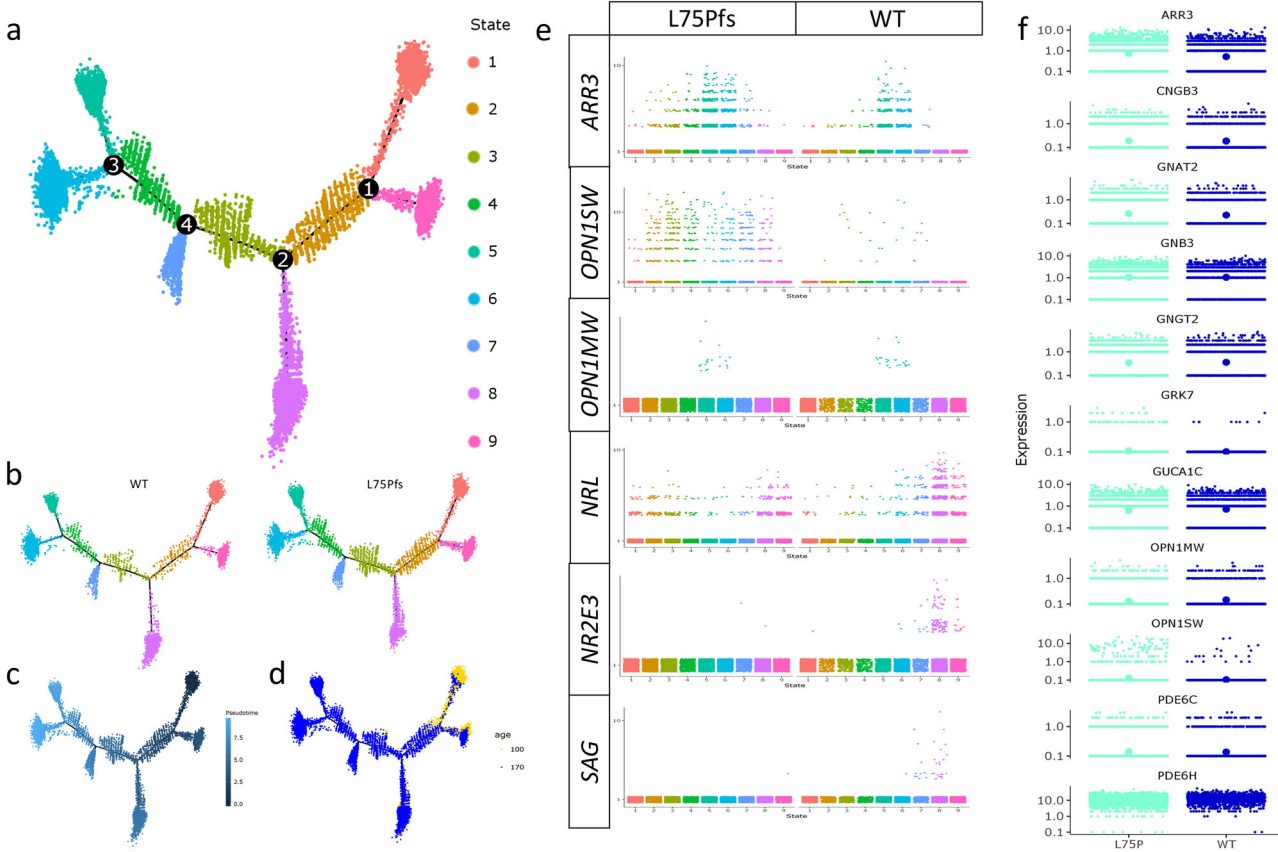

**Fig. 6 Combined analysis of WT and L75Pfs photoreceptor populations. a–d** Trajectory of combined WT and L75Pfs photoreceptors colored by state (**a**), genotype (**b**), pseudotime (**c**), or age (**d**). **e** Expression of rod and cone marker genes by state, with L75Pfs cell populations on the left and WT cell populations on the right. WT expression of *NR2E3* and *SAG* was used to assign states 7 and 8 as rods, and expression of *OPN1MW* to assign states 5 and 6 as cones. **f** Expression levels of cone markers in states 5 and 6 by genotype, with the position of the largest dot indicating the average level for each marker. All genes are expressed at comparable levels except *ARR3, GRK7, OPN1SW*, and *PDE6H*.

and validate its utility for analysis of perturbations occurring in NRL null photoreceptors.

**Reconstruction of the combined WT and L75Pfs trajectory.** After creating a WT trajectory that accurately represented photoreceptor development, we applied the same parameters to create a trajectory of 13,317 combined WT and L75Pfs photoreceptors to elucidate the shift in development resulting from the absence of NRL (Fig. 6a–d). Again, there were few contaminating bipolar cell precursors (254/13,317 cells with *VSX1* or *VSX2* expression) that likely did not affect trajectory reconstruction. The combined trajectory indicated nine cell states, compared to three states of the WT-only trajectory. We sought to characterize these states by their gene expression and gene ontology (GO). After the start of pseudotime with state 1, the first node separated two populations of immature photoreceptors. To characterize the photoreceptors in state 9, we input genes enriched in this state to a GO tool[41]. The enriched GO terms included various cell differentiation and cell stress processes (Supplementary Fig. 14). The remaining portions of the trajectory included developing photoreceptors (states 2, 3, and 4), and four branches corresponding to 2 rod and 2 cone cell fates. States 2, 3, and 4 were defined as developing photoreceptors due to expression of *CRX, OTX2*, and *RCVRN* and absence of WT cells with substantial rod or cone gene expression patterns (Supplementary Fig. 15b–d). These states have some cells expressing *NRL* or *ARR3*, but the low numbers and level of expression, compared to the more clearly-defined rod and cone populations of states 5, 6, 7, and 8, suggests

they are developing photoreceptors (Fig. 6e). Expression of *NR2E3* and *SAG* in WT cells identified states 7 and 8 as rod/rod-like cell fates and *OPN1MW* expression defined states 5 and 6 as cone fates. To differentiate between cone states 5 and 6, we utilized GO analysis on gene sets enriched in each fate (Supplementary Fig. 15e). State 5 cones were enriched for GO terms relating to electron transport chain and guanylate cyclase activity, whereas state 6 cones had enrichment for retinal development and photoreceptor differentiation terms, suggesting that state 6 cones may be less mature than the more metabolically active cones of state 5. Despite apparent differences in maturity and metabolism, cells in these states were combined into a single cone population for downstream analyses due to confidence in their identity as cone photoreceptors. We performed similar analyses on states 7 and 8 to distinguish between the two rod/rod-like populations (Supplementary Fig. 15f). While both populations were enriched for terms related to photoreceptor activity, state 8 cells appear to have stronger rod profiles due to enrichment in rhodopsin signaling terms. As with the cone states, cells of these two rod states were combined into one rod population for downstream analysis. This analysis identified the L75Pfs photoreceptor populations and their WT counterparts, thus allowing for further comparison of mature L75Pfs photoreceptor populations to WT rods and cones.

**Characterization of L75Pfs S-opsin expressing cells.** Because previous murine studies described Nrl−/− photoreceptors as possible "cods", we sought to determine if this would hold for

human NRL null photoreceptors[3]. To better characterize the L75Pfs photoreceptors, we determined genes differentially expressed compared to WT cells in each photoreceptor subset. Comparison of genes enriched in either WT or L75Pfs cone states (combined states 5 and 6) revealed 184 genes, few of which were rod- or cone-specific genes (Supplementary Data 5). Importantly, nearly all cone-specific genes showed no significant differential expression between WT and L75Pfs cones (Fig. 6f). Exceptions included *ARR3, GRK7,* and *OPN1SW,* which were more highly expressed in L75Pfs cells, and *PDE6H,* which showed slight enrichment in WT cells. Lower *OPN1SW* expression in WT cones was expected as ML-cones are the dominant cone subtype in WT organoids (Fig. 2p). The low number of differentially expressed genes and comparable expression levels of most cone genes between WT and L75Pfs cones suggests that this population of L75Pfs cones is essentially normal.

To characterize L75Pfs photoreceptors in the rod/rod-like branches (states 7 and 8), we compared them to both WT rods and WT cones by performing differential gene expression analysis according to the schematic in Fig. 7a (L75Pfs rod-like red cells vs WT rod purple cells; L75Pfs rod-like red cells vs WT cone blue cells). L75Pfs photoreceptors of the rod-like states had high *OPN1SW* expression, while also exhibiting WT rod levels of *OPN1MW* expression and WT cone levels of *NR2E3/SAG* expression (Fig. 7b). In comparing L75Pfs and WT rods/rod-like cells, 397 genes were differentially expressed (Supplementary

Data 6), with L75Pfs photoreceptors expressing significantly lower levels of the rod-specific genes *CNGB1, GNAT1, GNB1, GNGT1, NR2E3, NRL, PDE6G, ROM1,* and *SAG* than their WT counterparts (Fig. 7c). The expression levels of these rod genes in L75Pfs rod-like cells were comparable to their expression levels in WT cones (Supplementary Fig. 16a). The low expression of these genes compared to WT rods is likely due to the loss of NRL. In comparing L75Pfs rod-like cells to WT cones, 791 genes were differentially expressed, including many cone-specific genes (Supplementary Data 7). Compared to WT cones, L75Pfs rod-like cells had significantly lower expression of *ARR3, CNGB3, GNAT2, GNB3, GNGT2, GUCA1C, PDE6C,* and *PDE6H* (Fig. 7d). Except for *ARR3* and *PDE6H,* all of these genes were expressed at comparable levels in L75Pfs rod-like cells and WT rods (Supplementary Fig. 16b). Interestingly, L75Pfs rod-like cells also had significantly higher expression of *GNGT1,* a rod transducin also associated with foveal cones[38]. The high *OPN1SW* expression, rod levels of expression of cone genes, cone levels of expression of rod genes, and degree of differential gene expression compared to WT rods and cones suggests that L75Pfs rod-like cells are human analogs of "cods".

**MEF2C as a candidate regulator of cone cell fate.** Because the cod expression pattern is not completely consistent with either rods or cones, the presence of this population suggests that NRL

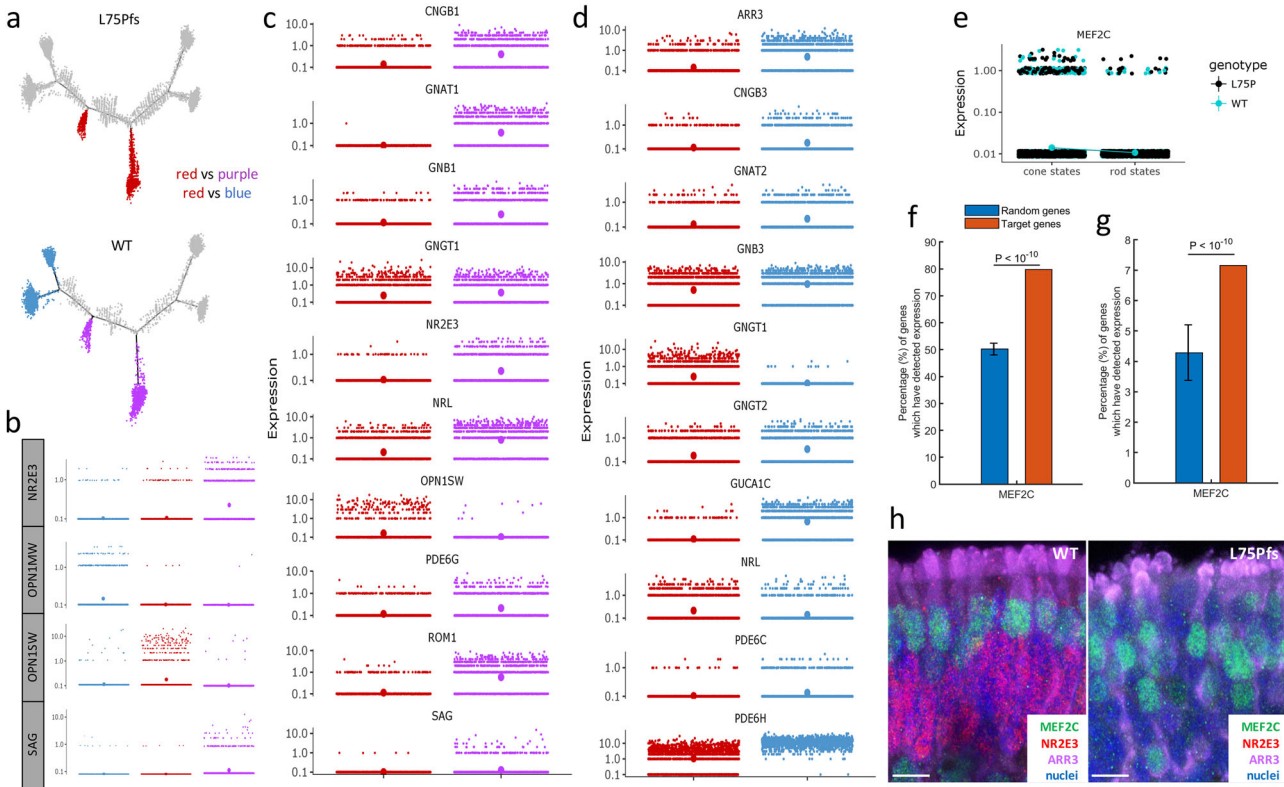

**Fig. 7 Comparative analysis of L75Pfs photoreceptors identifies 2 populations of S-opsin expressing photoreceptors, with one population having gene expression inconsistent with WT S-cones, and identification of MEF2C as a candidate regulator of cone cell fate specification. a** Plot depicting the differential expression analysis. Gene expression in the cells highlighted in red (L75Pfs rod-like cells) was separately compared to expression in the cells highlighted in blue (WT cones) and the cells highlighted in purple (WT rods). **b** Expression levels of *NR2E3, OPN1MW, OPN1SW,* and *SAG* across the three cell groups (red, blue, and purple) used for differential analysis. **c, d** Expression levels of rod- and cone-specific genes differentially expressed between L75Pfs rod-like cells and WT rods (**c**), or WT cones (**d**). The position of the largest dots indicate average gene expression levels for **b–e**. Expression level of *MEF2C* in cells in the cone states (5 and 6) on the left versus the rod states (7 and 8) on the right, with cells colored by genotype (black = L75Pfs, teal = WT). **f, g** Percentage of *MEF2C*-regulated genes expressed in a minimum of 10 cells (**f**) and differentially expressed between the red (L75Pfs rod-like cells) and blue (WT cones) groups (**g**) versus random genes. **h** Confocal images demonstrating MEF2C (green) co-expression with ARR3 (purple) in cone photoreceptors of WT (left) and L75Pfs (right) organoids at d160. NR2E3+ nuclei (red) are present in WT but not in L75Pfs organoids. Scale bar = 10 μm.

is not the only transcription factor directing rod versus cone photoreceptor fate specification. We postulated that other transcription factor(s) with higher expression in cones than L75Pfs cods could play a previously unappreciated role in regulating cone development. We identified 75 transcription factors of the 791 genes differentially expressed between WT cones and L75Pfs cods, with *MEF2C* as a potential candidate due to its higher expression level in both WT and L75Pfs cones compared to L75Pfs cods and WT rods (Fig. 7e). Interestingly, MEF2C has been shown to act downstream of *Nrl* in mice; however, in our dataset *MEF2C* showed significant enrichment in cone photoreceptors, suggesting a potential difference between human and murine photoreceptor fate determination[42]. To determine if *MEF2C* could be involved in human photoreceptor gene regulation, we queried human retinal ATAC-seq data for regions of open chromatin with MEF2C binding sites[43]. After identifying 475 genes with MEF2C binding sites, we tested for enrichment of MEF2C-regulated genes in our dataset. Three hundred and seventy five genes (80%) were expressed in at least ten cells in this photoreceptor dataset, a significant enrichment compared to the expected 50% for random genes (Fig. 7f). Furthermore, 34 genes (7.2%) were differentially expressed between L75Pfs cods and WT cone cells, a significant enrichment compared to 4.2% expected for random genes (Fig. 7g). This enrichment in expressed genes and differentially expressed genes suggests a possible role for MEF2C in photoreceptor development, and specifically rod versus cone cell fate specification. Consistent with our transcriptome data, MEF2C protein was detected exclusively in ARR3+ photoreceptors in both WT and L75Pfs retinal organoids at d160 (Fig. 7h, Supplementary Fig. 17a–j). MEF2C expression was also detected in d122 human fetal retina within a single-cell layer adjacent to NR2E3+ developing rod nuclei, consistent with cone localization (Supplementary Fig. 17k–n). Strong MEF2C expression was also detected in peripheral adult monkey retina in infrequent cells next to NR2E3+ rod nuclei, also corresponding to cones (Supplementary Fig. 17o–r), and within a line of INL nuclei likely corresponding to Müller glia. Additionally, faint expression was detectable in rod nuclei (Supplementary Fig. 17q). Furthermore, analysis of published adult human scRNAseq data confirmed enrichment of *MEF2C* expression in cones relative to rods (p = 0.04489, one-sided T-test)[24]. This protein localization profile, adult human photoreceptor expression analysis, and enrichment both in expressed genes and differentially expressed genes in our dataset suggests a possible role for *MEF2C* in human cone photoreceptor development or maturation.

## Discussion

We have used an in vitro human stem cell-based system to investigate photoreceptor development in *NRL* null human retinal organoids. L75Pfs has been identified as a null allele, and L75Pfs-derived retinal organoids have a phenotype similar to the phenotype of the *Nrl* null mouse[3,4,33]. Rods were not detected by immunocytochemistry (Figs. 1 and 2) and the portion of the ONL typically populated by rods in WT organoids instead contained S-opsin+ photoreceptors. This perturbation of photoreceptor cell fate determination was accompanied by a dramatic shift in the ratio of S- to ML-opsin+ photoreceptors (Fig. 2). Additionally, we confirmed that this phenotype was due to lack of NRL by reintroducing functional *NRL* and demonstrating restriction of S-opsin expression and promotion of RHO expression in mutant photoreceptors (Fig. 3).

To better understand the identity of the photoreceptor populations that develop in the absence of NRL, we performed scRNAseq to characterize the cell populations of WT organoids compared to L75Pfs organoids. Previous studies have analyzed

retinal organoid development using scRNAseq; however, the findings were limited by an inability to resolve cell clusters, the loss of INL cell populations, or incomplete characterization of photoreceptor populations[26–28]. Using transcriptomics, our analyses verified that retinal organoids generate all major neuronal cell types present in in vivo retina (Fig. 4). Our findings also enabled comparison of the timeline of retinal organoid and in vivo fetal retinal development. Previous fetal retinal bulk RNAseq analysis characterized three "epochs" of retinal development, with photoreceptor development occurring during the third epoch[44]. In comparing the timeline of retinal organoid development to that of the fetal retina, the organoids used in this study corresponded to this final epoch. Similar to in vivo retinal development, retinal organoids exhibit an early emergence of horizontal and amacrine cells, expression of cone-opsin prior to rhodopsin, and the later development of bipolar cells. Through our analyses we have generated a more comprehensive picture of the transcriptome of retinal organoids and provided a platform to further "stage" in vitro retinal organoid development compared to that of in vivo fetal retinas[19].

In addition to characterizing retinal organoid development in a WT system, we also provided a comprehensive picture of human photoreceptor development in the absence of NRL. Similar to a recent study characterizing cone photoreceptor development in human retinal organoids in the absence of the transcription factor THRB, we determined how loss of the rod transcription factor, NRL, affects rod and cone photoreceptor genesis[20]. While Nrl loss has been studied extensively in mice, to our knowledge this work represents the first transcriptomic characterization of human NRL null photoreceptors, and demonstrates similarities and differences between murine and human retinal development[3,30,33,45,46]. In both humans and mice, loss of NRL leads to development of an S-opsin+ photoreceptor dominant retina at the expense of rods. Murine Nrl null photoreceptors exhibited absent or decreased expression of *Rho, Gnat1, Nr2e3, Rom1, Rcvrn, Gnb1*, and *Cnga1*, consistent with our observations of decreased expression of these genes in human retinal organoids[3,45]. While *Sag* is expressed in murine Nrl null photoreceptors, albeit at lower levels, we detected no *SAG* expression in human NRL null photoreceptors (Fig. 6e). Human NRL null photoreceptors also exhibited substantial expression of the rod transducin *GNGT1*, which is not expressed in Nrl null mice. Consistent with upregulation of cone-specific genes in murine Nrl null photoreceptors, we also detected higher expression of *OPN1SW, GNAT2, GNGT2*, and *GNB3* in NRL null photoreceptors compared to WT. However, studies of Nrl null mice utilized population transcriptomics and thus only characterized the expression pattern of the average Nrl null photoreceptor[30]. Through single-cell analysis, we identified two distinct populations of NRL null photoreceptors, with one population bioinformatically indistinguishable from WT cones and the other representing a cod population with high *OPN1SW* expression but significantly lower expression of other rod- or cone- specific genes. This finding differs from previous studies that characterized Nrl null photoreceptors as "normal" S-cones[3,45]. Our model of two distinct *OPN1SW* expressing populations is compatible with the phenotype of enhanced S-cone syndrome due to the preservation of cone function in concert with retinal degeneration[4]. It is possible that preservation of S-cone function stems from the "normal" NRL null S-cone population, whereas the cod population may explain the degenerative phenotype. Of note, patients with enhanced S-cone syndrome lack normal retinal architecture, a phenotype not recapitulated in NRL null retinal organoids, and this disorganization may contribute to the degeneration observed in vivo[4].

In addition to characterizing photoreceptor populations, the availability of NRL null retinal organoids provided a platform for novel transcription factor discovery. Kim et al. combined transcriptomics and epigenetics in the context of Nrl null mice to identify candidate regulators of photoreceptor cell fate[30]. We utilized a similar methodology to identify *MEF2C* as a candidate regulator of cone cell fate specification due to its higher expression in cones and enrichment of differentially expressed genes with MEF2C binding sites. Interestingly, *MEF2C* had been identified through analyses of Nrl null mice, but as a transcription factor involved in conferring rod identity[42]. These opposing roles may be due to differences in human and murine retinal development, the existence of MEF2C splice variants, or the temporal expression pattern of *MEF2C* during retinogenesis, as Hao et al. detected *MEF2C* exclusively in mature murine retinas and not during development[42]. In both developing organoids and fetal human retina, MEF2C was primarily detected in cones and remained cone-enriched in adult primate retina (Fig. 7h, Supplementary Fig. 17). While our immunocytochemical evidence of MEF2C enrichment in developing cones supports a possible role for MEF2C in cone development, further studies, such as cell type-specific gain- and loss-of function experiments, will be necessary to determine experimentally the possible role of *MEF2C* in cone cell fate specification and/or maturation.

## Methods

**Generation of iPSC lines**. Tissue samples were obtained with written informed consent in adherence with the Declaration of Helsinki and with approval from institutional review boards at the University of Wisconsin-Madison and Massachusetts Eye and Ear Infirmary. The patient received clinical ophthalmic care from Dr. Eric Pierce in the ophthalmology clinic at the Children's Hospital of Philadelphia. The patient was recruited by Dr. Pierce to participate in research on the genetics of inherited retinal diseases. Participation involved consenting to collection of both blood and tissue samples. Informed consent was obtained from all donors of cells and tissues. A fibroblast biopsy from patient OGI-019-047 was reprogrammed via Sendai virus delivery of *OCT3/4, KLF4, SOX2,* and *cMYC* by the Waisman Center iPSC core (University of Wisconsin-Madison)[32]. The patient mutation (*NRL* c.233dup(C),p.(L75Pfs19X)) was confirmed in iPSCs and organoids by PCR amplifying a 770 nt genomic region surrounding the duplication with F-5′-TCCCTGCTCCTGGTTC-3′ and R-5′-CACCATCCCTCTGGCTTTCC-3′ followed by Sanger sequencing (University of Wisconsin Biotech Center, Madison, WI) with the F primer. Karyotype analysis was performed on iPSCs by WiCell (Madison, WI).

**Cell lines and retinal differentiation**. Three independent clones were used for all experiments and compared to three WT lines: WA09 (WiCell), 1013 and 1581[19]. All plasticware and reagents unless otherwise stated, were from ThermoFisher. All hPSCs were maintained on Matrigel in either mTeSR1 (WiCell) or StemFlex and passaged with either Versene or ReLeSR (STEMCELL Technologies). Retinal differentiation has been described[19]. Briefly, embryoid bodies (EB) were lifted with either 2 mg/ml dispase or ReLeSR and weaned into Neural Induction Media (NIM: DMEM:F12 1:1, 1% N2 supplement, 1× MEM nonessential amino acids (MEM NEAA), 1× GlutaMAX and 2 μg/ml heparin (Sigma)) over the course of 4 days. On day 6 (d6), 1.5 nM BMP4 (R&D Systems) was added to fresh NIM and on d7, EBs were plated on Matrigel at a density of 200 EBs per well of a 6-well plate. Half the media was replaced with fresh NIM on d9, d12 and d15 to gradually dilute the BMP4 and on d16, the media was changed to Retinal Differentiation Media (RDM: DMEM:F12 3:1, 2% B27 supplement, MEM NEAA, 1× antibiotic, anti-mycotic and 1× GlutaMAX). On d25–30, optic vesicle-like structures were manually dissected and maintained as free floating organoids in poly HEMA (Sigma)-coated flasks with twice weekly feeding of 3D-RDM (RDM +5% FBS (WiCell), 100 μM taurine (Sigma) and 1:1000 chemically defined lipid supplement) to which 1 μM all-trans retinoic acid (Sigma) was added until d100. Live cultures were imaged on a Nikon Ts2-FL equipped with a DS-fi3 camera or on a Nikon Ts100 equipped with a qImaging CE CCD camera.

**Immunocytochemistry and transmission electron microscopy**. Human prenatal eyes were obtained from the Laboratory of Developmental Biology (University of Washington-Seattle). Tissue collection methods adhered to Institutional Review Board requirements, NIH guidelines and the Helsinki declaration. Organoids were fixed in 4% paraformaldehyde at room temperature for 40 min, cryopreserved in 15% sucrose followed by equilibration in 30% sucrose, and sectioned on a cryostat. Slides were blocked for 1 h at RT in 10% normal donkey serum, 5% BSA, 1% fish gelatin and 0.5% Triton then incubated overnight at 4 °C with primary antibodies

diluted in block. Supplementary Table 2 lists primary antibodies, dilutions and sources. Slides were incubated with species-specific fluorophore-conjugated secondary antibodies diluted 1:500 in block, for 30 min in the dark at RT (Alexa Fluor 488, AF546 and AF647). For co-visualization of ML and S opsin, the rabbit α-ML opsin antibody (Millipore) was directly conjugated to AF555 using the Apex Alexa Fluor 555 labeling kit (ThermoFisher) according to manufacturer's instructions, mixed with unlabeled rabbit α-S opsin (Millipore) and detected with an anti-rabbit AF488 secondary. Sections were imaged on a Nikon A1R-HD laser scanning confocal microscope. Cone and rod counts were performed on at least six random images from at least 3 different organoids from each of the 3 WT lines and each of the three L75Pfs clones using Nikon Elements D annotations and measurements module. *P*-values were calculated with an unpaired two-tailed Student's *t*-test (Mann–Whitney test) using Graph-Pad Prism 6.

Organoids were fixed for TEM in 3% glutaraldehyde and 1% paraformaldehyde in 0.08 M sodium cacodylate buffer (all from Electron Microscopy Sciences) overnight with gentle rocking at 4 °C, washed with 0.1 M cacodylate buffer and post-fixed in 1% Osmium Tetroxide for 2 h at RT. The organoids were dehydrated in a graded ethanol series, further dehydrated in propylene oxide and embedded in Epon epoxy resin. Ultra-thin sections were cut with a Leica EM UC6 Ultramicrotome and collected on pioloform-coated 1 hole slot grids (Ted Pella, Inc.). Sections were contrasted with Reynolds lead citrate and 8% uranyl acetate in 50% EtOH and imaged on a Philips CM120 electron microscope equipped with an AMT BioSprint side-mounted digital camera and AMT Capture Engine software.

**RT-qPCR**. Stage 2/3 organoids from two differentiations each from three WT and 3 L75Pfs clones were collected at d160 (when PRs were starting to mature) and RNA was extracted using the RNAeasy mini spin kit (Qiagen) according to manufacturer's instructions, including the optional DNAse step. 1 μg of RNA was reverse transcribed using the iScript cDNA synthesis kit (BioRad), diluted 1:10 and qPCR was performed with SSO Advanced Sybr Green Master mix (BioRad) on a StepOne Plus qPCR machine (ABI). A list of primers can be found in Supplementary Table 3. ΔC(q)s were calculated from the geometric mean of two housekeeping genes and variability for all six lines was visualized by subtracting the ΔC(q) for each differentiation from the avg ΔC(q) for all six WT samples. $2^{-\Delta\Delta C(q)}$ were plotted and *P*-values were calculated using an unpaired two-tailed Student's *t*-test (Mann–Whitney test) using Graph-Pad Prism 6.

**Lentiviral rescue**. Full length human NRL coding sequence was PCR-amplified from human adult neuroretina cDNA using F-5′-ATGGCCCTGCCCCCCAGC-3′ and R-5′-TCAGAGGAAGAGGTGGGAGGGGTC-3′ with a BamHI site added 5′ and a SalI site added 3′ to facilitate cloning into the pSIN-WP-mpgk lentiviral shuttle vector[47]. Lentivirus was produced via transfection of HEK293T cells, concentrated 40-fold by ultracentrifugation and titered on hiPSCs to calculate a working titer[48]. D90, early stage 2 organoids were infected every 3 days three times with effective titer to give >70% infection of iPSCs, by adding virus to the media. Control pgkGFP virus infection was monitored by live fluorescence. Organoids were maintained in culture for 100 days to allow sufficient time for RHO expression, whereupon they were fixed and processed for ICC as described above. The pgkNRL infection was repeated four times- once with each clone except for clone 7 which was infected twice.

**Single-cell dissociation and sample preparation**. At d100, cell lines used were 1013 and 1581 for WT and L75Pfs clones 9 and 11. D170 organoids were 1013 for WT and clones 7 and 9 for L75Pfs. Organoids were dissociated to single cells with papain (Worthington) to 1 mg/ml and 5 μL DNase (Roche) per mL, using 200 uL papain mix per organoid. After 1–2 h when organoids appeared fully dissociated the reaction was quenched with media containing 10% FBS (Gibco). Single cells were resuspended in HBSS (Gibco) and 0.1 mg/mL BSA at 120,000 cells per mL. Single-cell capture was performed using a home-made Dropseq setup according to the published Dropseq protocol[21]. Cells were combined in oil (Biorad) and barcoded beads (Chemgenes) in ~1 nL droplets. Droplets were broken using 6× SSC and perfluorooctanol (Sigma) to collect beads. Reverse transcription was performed using Maxima reverse transcriptase (Thermo Fisher Scientific) and cDNA was amplified using Kapa (Roche). cDNA was quantified using a Bioanalyzer DNA High Sensitivity Chip (Agilent). cDNA was fragmented and libraries were created using the Nextera XT library prep kit (Illumina). Libraries were quantified by Qubit dsDNA HS (Thermo Fisher Scientific) and sequenced via Illumina HiSeq 2500.

**Bioinformatics**. Fastq files were processed according to the published Dropseq analysis pipeline and aligned to GRCh38 to extract expression matrices. Canonical component analysis, tSNE, and clustering was performed using the Seurat R package (version 2)[36]. To remove low quality cells, cells with less than 200 genes or >20% (d170) or 15% (d100) mitochondrial RNA content were filtered out. Genes expressed in a minimum of two cells were included for downstream analysis. The union of the top 2000 highly variable genes of WT and L75Pfs were used for canonical component analysis and 20 (d170) and 17 (d100) canonical components and were used to align subspaces and cluster cells. Cell populations were identified using previously published marker genes (Supplementary Table 1) and differential gene expression analysis was performed within each cluster to identify genes that

varied by genotype. Differential expression analysis within each population between WT and L75Pfs cells was performed using a Wilcoxon rank sum test. Bonferroni corrected p-values were calculated using the number of genes expressed at each time point (23645 for d100 and 25622 for d170). Cell populations at each age were compared to published fetal retinal cell populations and adult peripheral versus foveal populations[23,24]. For each population, the average expression of each gene expressed in both datasets was input for correlation analysis. Spearman correlation tests were performed for each cell population against its corresponding fetal and adult populations. Photoreceptor clusters of d100 and d170 were sub-setted out for analysis using the Monocle R package (version 2)[39]. A semi-supervised differential gene expression test was used on WT cells to identify genes that varied by age and cell type (NR2E3 = rod, ARR3 = cone, neither = undifferentiated photoreceptor). The top 780 non ribosomal or mitochondrial genes were used for ordering and trajectory reconstruction for both the WT trajectory and combined trajectory. Differential gene expression analysis was performed on the node in the WT trajectory to identify genes with rod/cone branch dependent expression. Similar analyses were performed on published adult photoreceptors[24]. Peripheral and foveal photoreceptors combined, and rod/cone designations were retained for the analysis. A differential gene expression test was performed to identify genes differentially expressed across cell types, and genes with $q$ values < 0.00001 (602 genes) were used for trajectory reconstruction. Additional differential gene expression tests were performed on subsets of the merged trajectory to characterize L75Pfs photoreceptors compared to WT rods and cones. All differential gene expression analysis in Monocle was performed using a likelihood ratio test on the 14,657 genes expressed in a minimum of 10 cells and a Bonferroni corrected p-value. To analyze possible regulation by MEF2C, human retinal ATAC-seq data was queried for genes with MEF2C binding sites[43]. The resulting gene list (475 genes with MEF2C binding sites) was compared against genes expressed in at least 10 cells of the photoreceptor dataset (14,657 genes) and against genes differentially expressed (791 genes) between the WT cones and L75Pfs rod-like cells. The percentage of expressed genes and differentially expressed genes with MEF2C binding sites were compared against the expected percentages using a two-tailed Student's $T$ test. To estimate expected frequency in expressed genes, 14,657 genes were randomly chosen from all 28,040 human genes and the number of genes with MEF2C binding sites was determined and this process was repeated for 1000 iterations. This process was repeated for differentially expressed genes, randomly choosing 791 genes of the 14,657 expressed genes to determine the expected frequency of differentially expressed genes with MEF2C binding sites.

**Statistics and reproducibility**. Statistics for immunocytochemical analyses and RT-qPCR was calculated using two-tailed Students $T$-tests and scRNAseq statistics were computed using the built-in statistical tests in the Seurat and Monocle R packages, specifically Wilcoxon Rank Sum tests and Likelihood Ratio tests. Replicates were defined as individual organoids, with a minimum of 2–3 organoids per group. When two replicates were used, two cell lines were used to ensure >3 replicates per condition. Reproducibility was confirmed by consistency across all replicates.

**Reporting summary**. Further information on research design is available in the Nature Research Reporting Summary linked to this article.

## Data availability

All raw sequencing files and expression matrices generated under this study can be found at GEO under the accession number: GSE143669. Raw data used to create Fig. 2o–q supplied as supplementary material.

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

## Acknowledgements

We are grateful to the Genetics Resources Core Facility (Johns Hopkins) and the Electron Microscope Facility, the Department of Dermatology lentivirus core, and the Waisman Center Intellectual and Developmental Disabilities Models core at the University of Wisconsin-Madison for assistance, and to Dr Celio Pouponnot (Institut Curie) for sharing unpublished NRL qPCR primer sequences. This work was supported by grants from Foundation Fighting Blindness, Research to Prevent Blindness, Maryland Stem Cell Research Fund, the Sandra Lemke Trout Chair in Eye Research, Retina Research Foundation Emmett Humble Distinguished Directorship, and the National Institutes of Health (P30 EY001765, U54 HD090256, K99 EY 0290112, and R01EY021218-06A1), and generous gifts from the Guerrieri Family Foundation.

## Author contributions

A.K. conceived and designed the study, conducted experiments, acquired and interpreted data, and wrote the manuscript. E.E.C. conceived and designed the study, conducted experiments, acquired and interpreted data, and wrote the manuscript. J.W. interpreted data. A.D.J. and K.L.E. acquired and interpreted data. A.M.K., L.C., and T.-H.W. established the DropSeq platform in the lab. C.B. conceived and designed the study, edited the manuscript, and provided funding support. M.J.P. conceived and designed the study and interpreted data. E.A.P. provided the patient iPSC line. J.Q. interpreted data. D.M.G. conceived and designed the study, interpreted data, edited the manuscript, and provided funding support. D.J.Z. conceived and designed the study, edited the manuscript, and provided funding support.

## Competing interests

D.M.G. and M.J.P. have an ownership interest in Opsis Therapeutics LLC, which has licensed the technology to generate optic vesicles from pluripotent stem cell sources reported in this publication. D.M.G. also declared intellectual rights through the Wisconsin Alumni Research Foundation and a consultant role with FUJIFILM Cellular Dynamics International. All other authors indicated no potential conflicts of interest.
