## [Peer Review File · Communications Biology]

Reviewers' comments:

Reviewer #1 (Remarks to the Author):

In the manuscript titled "INVESTIGATING CONE PHOTORECEPTOR DEVELOPMENT USING 2 PATIENT-DERIVED NRL NULL RETINAL ORGANIDS" the authors present immunocytochemical and single cell RNAseq data describing the role of NRL loss in normal photoreceptor cell development and disease. In general, the manuscript is well written and presents an extensive body of very convincing high quality data, that will be of interest to a broad readership.

To strengthen the manuscript, the following minor revisions are suggested.

Introduction

Line 63-73: Please add (PMID 29385733) along with reference #4 to support the statement that recessive mutations in NRL cause Enhanced S Cone Syndrome, i.e. NRL is a very rare cause of this disease, typically mutations in NR2E3 are the culprit. As NRL variants cause a range of clinical phenotypes, inclusion of a sentence or two more accurately depicting the genotype phenotype relationship would be useful, i.e., mutations in NRL cause clinical phenotypes ranging from x-y. This is important as typically, homozygous null NRL mutations as the patient in this study have, would cause a very severe early onset phenotype that may not be called Enhanced S Cone Syndrome clinically. Along these lines, inclusion of the clinical diagnosis of the iPSC donor used in this study would be useful, i.e. how was their disease described clinically?

Lines 78-79: In addition to the mouse study referenced, single cell RNAseq has been performed on both developing (PMID 31269016) and normal adult human donor retina (PMIDs 31075224, 31436334). In these studies, analysis of retinal cell populations etc., was been described, please reference.

Results

Figures 1 and 2, Lines 105-111: Description of the developmental phenotype presented in figures 1 and 2 has been largely been previously described by the group, i.e. reference #17. The reviewer recommends inclusion of the replicate data as either supplemental or a reference, i.e. "we have previously shown" and novel data in the main figures.

Figure 5, Lines 235-241: Comparison to previously published single cell RNAseq of human developing and mature retina, see suggested references above, should be performed. Specifically, how comparable are the normal developmental trajectories and how similar are the cells in organoids to mature peripheral vs macular cells.

Discussion:

While the reviewer agrees with the authors that it "is possible that the preservation of S-cone function stems from the "normal" NRL null S-cone population, whereas the cod population may explain the degenerative phenotype.", it should also be pointed out that patients with enhanced S cone syndrome do not have normal retinal structure, i.e. disorganized retinas with tubulations etc. As such the degenerative phenotype may be more of a by standard effect. As per the introduction, further discussion of the range of clinical phenotypes caused by NRL would be useful in this section.

Line 416: The statement "Because the retinal organoids do not represent a final stage of maturity, it is possible that MEF2C may show expression in rods at more mature developmental time points." Presents an interesting hypothesis that could readily be confirmed by evaluating published single cell RNAseq data sets from developing and mature human retina.

Reviewer #2 (Remarks to the Author):

This study examines the histological and molecular alterations that occur in the presence or absence of NRL, using human PSC-derived retinal organoids as a model system. The authors demonstrate a clear phenotype in the retinal organoids, derived from 3 iPSC clones (L75Pfs) generated from an NRL null patient. They determine an increased percentage of S OPSIN+ cone photoreceptors, present in the L75Pfs cell lines compared to normal controls, as well as a lack of both NRL and NR2E3 protein in the retinal organoids. This follows what has been observed in patients, who demonstrate a reduced number of rod photoreceptors and increased short-wavelength sensitive S cones. This phenotype could be rescued by the introduction of functional NRL, using lentiviral transduction of the patient organoids and demonstrated by the appearance of RHODOPSIN+/ARRESTIN3- photoreceptors. The authors then utilised single-cell RNA sequencing to further examine the discrete photoreceptor sub-populations present in NRL null organoids. They determined the presence of two populations of S-OPSIN expressing cones, one typical of normal cones and one reminiscent of the cod population (rod/cone hybrid photoreceptors) observed in the *Nrl*^{-/-} mouse model. Finally, by examining the differential expression of transcription factors between the normal cones and the L75Pfs rod/cone intermediates, they identified a novel regulator of human cone photoreceptor development. This study demonstrates the potential for elucidating not only human disease phenotypes but also developmental processes, using patient iPSC lines as a platform.

There are a few minor points that would improve the clarity of the manuscript for the reader and a couple of typographical errors that are listed below.

1. In supplementary figure 1 the authors show the karyotype analysis of all three iPSC clones generated, as well as the presence of the c.223dupC mutation in NRL. However, they do not comment on or show any images of the morphology of the iPSCs or any characterisation of pluripotency markers present in these cells. These are standard criteria for confirming the generation of iPSC lines and should be included, as it is not stated that these lines have been published previously.
2. The authors refer to the retinal organoids examined in figure 2 as being stage 3, in line with their previously published staging system. However, it is unclear the criteria they use to determine this, as in the text (page 5, line 109) they state 'as photoreceptors matured and formed outer segments' but they show no data to support the formation of outer segments in these organoids. Do they instead mean the presence of segment-like structures or a brush-border as observed by light microscopy in fig 2c and 2i? If so, I think this should be clarified for the reader or alternatively outer segment-specific markers, such as PERIPHERIN or ABCA4 shown.
3. Despite being described in the methods (page 21, line 474), no statistical tests are stated in the main text or on figure 2O and 2P. In addition, the authors do not comment on the reduced percentage of photoreceptors in the ONL of the L75Pfs organoids (Fig 2O), is there a population of ARR3-/NR2E3- photoreceptors? It would be useful for the reader to discuss the reason for this disparity in comparison with the normal WT organoids.
4. On page 6, line 126, the authors state that L75Pfs organoids did not display increased rosette formation or demonstrate a disrupted OLM, as opposed to the *Nrl*^{-/-} mouse. However, rosette formation in this mouse model has been shown to depend on 11-cis-retinal, as the absence or reduced activity of RPE65 in double-mutant mice prevents rosette formation (Wenzel A et al. IOVS 2007; Samardzija M et al. IOVS 2014). Therefore, it is unlikely to result from the constraints of the developing eye in vivo.
5. On page 7 (line 148) the authors state that RHODOPSIN+ cells were not observed in untreated L75Pfs organoids but do not mention pkgGFP control treated organoids? An image of Rhodopsin staining in the control treated organoids is not shown in figure 3 and is required to compare with Fig 3S, as has been included for NRL in Fig 3g. In addition, as RHODOPSIN+ cells in the WT organoids are

not shown elsewhere in the paper it would benefit the reader if an image was included somewhere, to show the expected localisation of RHODOPSIN at this stage of development.

6. On page 8 (line 173) the authors mention that more mature retinal cell type populations were captured at the later day 170 time point, but the loss of the retinal ganglion cell population should also be noted at this later stage.

7. For Figure 6, images B-D and F should be referred to in the main text.

8. On page 15 (line 341), Supplementary Figure 11A-J should be stated, rather than 10A-I and on line 344, Supplementary Figure 11K-N should be referred to, instead of 10J-M. Again, on page 19 (line 419) Supplementary Figure 11 should be noted, not 10.

9. Finally, the presence of MEF2C in cone photoreceptors in retinal organoids at day 160 and day 122 in human fetal retina does suggest a possible role in cone development. As it appears to be present in adult rod photoreceptors in mice, have the authors examined any later stage organoids (>200 days) for the presence and localisation of MEF2C, or has it been examined in adult human retinal sections at all?

This is a detailed and thorough study of the cone photoreceptor populations present in NRL null human retinal organoids that is well presented and of interest to the fields of both retinal development and disease. Two other studies have examined the cell populations present in human retinal organoids over time using single-cell RNA sequencing, however, this study is the first to examine discrete photoreceptor sub-populations together with a genetic aberration. Overall, this is an original and important study that would be of significant interest to a wide audience and is highly recommended for publication in Communications Biology.

Reviewer #3 (Remarks to the Author):

This paper by Zack, Gamm and colleagues explores the role of the transcription NRL in the specification of photoreceptor fate. The authors use induced Pluripotential Stem Cells from normal subjects and a patient with a null mutation in NRL to generate retinal organoids using previously published methods. This results in the formation of in vitro retinal structures that bear reasonably well laminated neural retina that exhibit all 7 retinal cell types when derived from normal stem cells. In contrast, those generated from the NRL-null iPSCs yield retinal structures that are devoid of rod photoreceptors, presenting instead with an excess of cone photoreceptors. This mimics what is seen clinically, where patients present with "enhanced S cone syndrome".

There is extensive literature on the role of Nrl in the mouse retina and how, in its absence, photoreceptors are diverted to an S cone fate. Other recent reports have also begun to describe the role of NRL in the human retina. However, the findings presented here shed important light on to the nature of the photoreceptors generated. Previous reports in the mouse have asserted that the cone-like cells generated in the absence of Nrl are true cones, rather than a hybrid state, as had been postulated in the early studies. Here, Zack and colleagues show that the original idea was most likely correct – that the cells destined to become rods, in the absence of NRL, instead adopt a hybrid state that is similar but not identical to an S cone photoreceptor. Alongside this, cells originally destined to become cones likely continue to form true cones.

These findings provide important verification in human retina of the role of NRL and the downstream network in the specification of rods. They also confirm an earlier, but subsequently neglected, hypothesis that photoreceptors lacking Nrl form a hybrid state, rather than true cones. Moreover, they identify new candidates that may play important roles in this specification process.

Comments:

1) The assertion that reintroduction of NRL into the organoids restores rod formation is overstated. It does certainly reduce S OPSIN expression in those cells, but RHODOPSIN expression is negligible (only one convincingly RHO+ cell is shown), and no other markers are assessed. The authors should qualify their conclusions and ideally provide further evidence of other early rod markers following the introduction of NRL. It is not unreasonable that forced expression of NRL only partially rescues the rod phenotype, but this should be made clear.

The staining for RHODOPSIN is nicely specific in 3m but is unconvincing in 3r-t and it is not possible to discern whether there is or is not colocalization with ARR3. Panel s appears to be at a lower magnification to r and t making the supposed colocalization difficult to assess. Better images and staining are required along with single confocal sections. In 3k, o I assume that NRL labelling is immunostaining (no note of a GFP reporter in the NRL lentiviral construct), in which case why is it not localized to the nucleus? Can the authors comment on why RHO is so low given the relatively robust clusters of NRL+ cells (in "o", at least). To what degree is RHO normally expressed in WT cultures of the same stage?

2) The authors note differences in the *Nrl*^{-/-} mouse retina and hPSC-derived organoids with respect to OLM integrity and rosette formation. They suggest that the hiPSC-derived retinas do not form rosette because they are not spatially constrained. They should also note the work by Grimm and colleagues on the role that RPE65 plays in the formation of whorls/OLM disturbances in the *Nrl*^{-/-}. This is of interest given the lack of significant amounts of RPE in the organoids.

3) P18 - The idea that the true cones survive, and the cods degenerate in the NRL null is particularly interesting. Is there evidence of degeneration in the L75Pfs organoids (caspase staining?), as might be predicted from the *Nrl*^{-/-} mice and the clinical phenotype. Related, can the authors comment on whether the MG in L75Pfs are in a more activated state compared to WT MG? Do the authors know if the true cones correctly locate to the apical most side or do they remain muddled with the cod like cells?

Minor:

4) P6, l147 – was ARR3 found in the NRL+ cells? Unclear from images.

5) P8, l178 – did the authors mean "incomplete"?

6) P10, l214 – sentence is a little ambiguous. Please clarify.

7) P22, l494 – reference to Supplementary Table "Y"?

8) Fig 2 – what is the red signal in 2a? What accounts for the missing 20% in the L75Pfs graphs?

9) Fig 4 – it would be helpful to have the cell types coloured the same in 4A and 4C

10) Fig 6 – please make the colour dots for the different states larger so it is easier to identify the different populations referred to in the text.

11) Figures, it would be helpful to state the days in culture (d100 or d170) on the images on the different EBs

12) Methods for counting cell number – the selection of 6 ROIs per organoid and 3 organoid per condition is quite limited given the established variability between EBs even within the same well. It is also not clear if these are made in a blinded manner. Although the reported magnitude of change is very large, and the conclusions are not in doubt, it would be better practice to implement a more robust sampling procedure for assessing no. of immuno-positive cells e.g. for Fig.2.

13) Supp table 6 – l154 has been autocorrected to a date

Point-by-point response to reviewers' comments
INVESTIGATING CONE PHOTORECEPTOR DEVELOPMENT USING
PATIENT-DERIVED NRL NULL RETINAL ORGANIDS
COMMSBIO-19-1176-T

Reviewer 1:

Introduction

Line 63-73: Please add (PMID 29385733) along with reference #4 to support the statement that recessive mutations in NRL cause Enhanced S Cone Syndrome, i.e. NRL is a very rare cause of this disease, typically mutations in NR2E3 are the culprit. As NRL variants cause a range of clinical phenotypes, inclusion of a sentence or two more accurately depicting the genotype phenotype relationship would be useful, i.e., mutations in NRL cause clinical phenotypes ranging from x-y. This is important as typically, homozygous null NRL mutations as the patient in this study have, would cause a very severe early onset phenotype that may not be called Enhanced S Cone Syndrome clinically. Along these lines, inclusion of the clinical diagnosis of the iPSC donor used in this study would be useful, i.e. how was their disease described clinically?

RESPONSE: We have added the mentioned reference and edited the text to read: “However, the range of clinical phenotypes caused by different *NRL* mutations is broad, with autosomal dominant missense mutations leading to a clinical picture more akin to retinitis pigmentosa⁴⁻⁶. Similarly, enhanced S-cone syndrome can be caused by mutations in genes other than *NRL*, most commonly *NR2E3*.” In terms of more information about the patient, we agree that more clinical information would be helpful. However, we have on several occasions tried to obtain more clinical information but for complicated reasons we have not been able to succeed in these efforts.

Lines 78-79: In addition to the mouse study referenced, single cell RNAseq has been performed on both developing (PMID 31269016) and normal adult human donor retina (PMIDs 31075224, 31436334). In these studies, analysis of retinal cell populations etc., was been described, please reference.

RESPONSE. We have added the mentioned references. Thanks for pointing these out, and our apologies for having missed them.

Results

Figures 1 and 2, Lines 105-111: Description of the developmental phenotype presented in figures 1 and 2 has been largely been previously described by the group, i.e. reference #17. The reviewer recommends inclusion of the replicate data as either supplemental or a reference, i.e. “we have previously shown” and novel data in the main figures.

RESPONSE. While we agree with the reviewer's point that the WT phenotypes presented in Figures 1 and 2 have been well described in reference 17, we feel that readers benefit from having the WT image present to compare to. Additionally, each lab's protocol varies a bit, so we

feel providing organoid data from our labs serves as a tighter control for our mutant data. Thus, we would prefer to leave the figures in their current location.

Figure 5, Lines 235-241: Comparison to previously published single cell RNAseq of human developing and mature retina, see suggested references above, should be performed. Specifically, how comparable are the normal developmental trajectories and how similar are the cells in organoids to mature peripheral vs macular cells.

RESPONSE. Thank you for this excellent suggestion. We have performed spearman correlations between the organoid cell populations and published fetal/adult peripheral vs macular data and added supplementary figures 7- 8 for d100 and 10-11 for d170, as well as the lines: “Spearman correlations were performed between each cell type and published fetal and adult retinal scRNAseq datasets. This analysis revealed d100 organoids yielded amacrine cells, horizontal cells, and retinal ganglion cells that were more similar to fetal retinal cell populations. Rods most closely resembled adult peripheral rods, while cones and muller glia more closely resembled adult foveal cells.”

We also added: “Again, spearman correlations were performed between each cell type and published fetal and adult retinal scRNAseq datasets (Supplementary Fig. 9)²³⁻²⁴. Like d100, organoids at d170 yielded amacrine cells and horizontal cells more similar to fetal cells. Rods and bipolar cells more closely resembled peripheral cells, and, similarly to d100, cones and Müller glia were more highly correlated with adult foveal cells.”

To provide the methods for this new analysis, we added the following to the methods: “Cell populations at each age were compared to published fetal retinal cell populations and adult peripheral versus foveal populations. For each population, the average expression of each gene expressed in both datasets was input for correlation analysis. Spearman correlation tests were performed for each cell population against its corresponding fetal and adult populations.

Regarding how comparable the trajectories are, we cannot comment on this point as the mentioned references either have not performed these analyses at all, or have not analyzed exclusively photoreceptors. We attempted to perform this analysis on fetal PRs but could not get a clean separation between rods and cones – possibly due to the low cell number (144) or their lack of maturity.” To further develop this suggestion, we checked the novel markers identified from the trajectory against published adult retinal data and used a 1-sided T test to confirm significant enrichment for MAP4 ($p=4.223e-15$), MYL4 ($p=8.744e-13$), MCF2 ($p=6.225e-9$), KIF2A ($p=0.0002736$) in cones and CABP5 ($p<2.2e-16$), IRX6 ($p=0.02313$) in rods. To describe this analysis to the manuscript, we added: “We then checked published adult human scRNAseq data and confirmed statistically significant enrichment of MAP4, MYL4, MCF2, and KIF2A in cones, and CABP5 and IRX6 in rods ($p<0.05$, one-sided T test).²⁴” (lines 385-387).

Discussion:

While the reviewer agrees with the authors that it “is possible that the preservation of S-cone function stems from the “normal” NRL null S-cone population, whereas the cod population may explain the degenerative phenotype.”, it should also be pointed out that patients with enhanced S cone syndrome do not have normal retinal structure, i.e. disorganized retinas with tubulations etc. As such the degenerative phenotype may be more of a by standard (SIC) effect. As per the introduction, further discussion of the range of clinical phenotypes caused by NRL would be

useful in this section.

RESPONSE. As noted above, we have added more information on the range of clinical phenotypes caused by NRL.

Line 416: The statement “Because the retinal organoids do not represent a final stage of maturity, it is possible that MEF2C may show expression in rods at more mature developmental time points.” Presents an interesting hypothesis that could readily be confirmed by evaluating published single cell RNAseq data sets from developing and mature human retina.

RESPONSE.

We have provided supporting evidence of MEF2C enrichment in cones via adult monkey retina ICC and analysis of adult human retina single cell RNAseq data indicating an enrichment of MEF2C in adult human cones.

Reviewer 2:

There are a few minor points that would improve the clarity of the manuscript for the reader and a couple of typographical errors that are listed below.

1. In supplementary figure 1 the authors show the karyotype analysis of all three iPSC clones generated, as well as the presence of the c.223dupC mutation in NRL. However, they do not comment on or show any images of the morphology of the iPSCs or any characterisation of pluripotency markers present in these cells. These are standard criteria for confirming the generation of iPSC lines and should be included, as it is not stated that these lines have been published previously.

RESPONSE. We have added panels I-T to supplementary Figure 1 to show iPSC morphology and immunochemistry for pluripotency factors in all 3 L75Pfs clones.

2. The authors refer to the retinal organoids examined in figure 2 as being stage 3, in line with their previously published staging system. However, it is unclear the criteria they use to determine this, as in the text (page 5, line 109) they state ‘as photoreceptors matured and formed outer segments’ but they show no data to support the formation of outer segments in these organoids. Do they instead mean the presence of segment-like structures or a brush-border as observed by light microscopy in fig 2c and 2i? If so, I think this should be clarified for the reader or alternatively outer segment-specific markers, such as PERIPHERIN or ABCA4 shown.

RESPONSE. We have modified Line 109 (now line 119) to read: “As photoreceptors matured and formed outer segments (visible as the “hair-like” projections in Figure 2c and 2i), L75Pfs organoids showed a striking S-opsin dominant photoreceptor phenotype (Fig. 2)¹⁹.”

3. Despite being described in the methods (page 21, line 474), no statistical tests are stated in the main text or on figure 2O and 2P. In addition, the authors do not comment on the reduced percentage of photoreceptors in the ONL of the L75Pfs organoids (Fig 2O), is there a population

of ARR3-/NR2E3- photoreceptors? It would be useful for the reader to discuss the reason for this disparity in comparison with the normal WT organoids.

RESPONSE. We have added description of statistical test in two places: to line 136 which now reads: "...significantly increased in the L75Pfs organoids (Fig. 2Q, $p < 0.005$, Mann-Whitney test)" and to line 788, which now reads: " $p < 0.005$, Mann-Whitney test." In terms of the reduced percentage of photoreceptors in the ONL of the L75Pfs organoids, we have added a discussion of the reduced percentage of photoreceptors in L75Pfs ONL to line 117-118 (now lines 124-132) which now reads: "Quantification of ARR3+ cones and NR2E3+ rods as a percent of total nuclei in the ONL of WT and L75Pfs organoids (Fig. 2D-H and J-N) revealed a dramatic reduction in rods and an increase in cones in the L75Pfs ONL (Fig. 2O). Interestingly, while WT had rare nuclei in the ONL that were negative for both ARR3 and NR2E3, ~20% of the L75Pfs ONL nuclei expressed neither marker. Since ARR3 is normally expressed >60 days after the first cone progenitors are detected, these ARR3-/NR2E3- cells may represent mutant rod cells that either have not committed to a cone fate or do not yet express ARR3. We also quantified the ML- or S-opsin expressing cells as a percentage of the total ARR3 expressing cells and detected a 38-fold shift in the ML:S-opsin cone ratio from 19:1 in WT to 1:2 in L75Pfs organoids (Fig. 2P). Additional analyses of expression of key rod and cone genes by RT-qPCR revealed that genes associated with rod development were downregulated in L75Pfs mutant organoids relative to WT organoids, while S opsin expression was significantly increased in the former (Fig. 2Q). Thus, in L75Pfs human retinal organoids, rods appeared to be shifted toward an S cone fate, consistent with the phenotype of the *Nrl*^{-/-} mouse^{3,28}."

4. On page 6, line 126, the authors state that L75Pfs organoids did not display increased rosette formation or demonstrate a disrupted OLM, as opposed to the Nrl^{-/-} mouse. However, rosette formation in this mouse model has been shown to depend on 11-cis-retinal, as the absence or reduced activity of RPE65 in double-mutant mice prevents rosette formation (Wenzel A et al. IOVS 2007; Samardzija M et al. IOVS 2014). Therefore, it is unlikely to result from the constraints of the developing eye in vivo.

RESPONSE. We agree with the reviewer's comment and have removed our comment referring to rosette formation (lines 129-131, now line 143).

5. On page 7 (line 148) the authors state that RHODOPSIN+ cells were not observed in untreated L75Pfs organoids but do not mention pkgGFP control treated organoids? An image of Rhodopsin staining in the control treated organoids is not shown in figure 3 and is required to compare with Fig 3S, as has been included for NRL in Fig 3g. In addition, as RHODOPSIN+ cells in the WT organoids are not shown elsewhere in the paper it would benefit the reader if an image was included somewhere, to show the expected localisation of RHODOPSIN at this stage of development.

RESPONSE. We apologize for this oversight and have added pkgGFP treated organoids to line 160. We have also added a new Supplementary Figure 4 showing ICC for ARR3 and RHO in a stage 3 WT organoid and added a reference to it in line 161-163. And finally, we have added new panels 3I-L to show RHO expression in pkgGFP organoids. Lines 159-131 have been modified to read: "Rare RHO+ cells (Fig 3Q and 3W), which were never found in untreated or

pgkGFP-transduced (Fig 3I-L) L75Pfs NRL organoids, were uniformly negative for S-opsin (Fig. 3T) or ARR3 (Fig.3X) upon immunostaining. Of note, RHO localized to outer segments in some cells (Fig. 3S) is reminiscent of immunostaining seen in WT organoids (Supplementary Figure 4A-D).”

6. On page 8 (line 173) the authors mention that more mature retinal cell type populations were captured at the later day 170 time point, but the loss of the retinal ganglion cell population should also be noted at this later stage.

RESPONSE. We have added a mention of the loss of RGCs and suggested possible explanations: “Loss of retinal ganglion cells was also observed at this age. Possible explanations for retinal ganglion cell loss include microfluidic bias favoring other cell types or death of retinal ganglion cells due to the lack of vasculature in retinal organoids. Notably, age-dependent RGC loss has been reported to be a normal phenomenon in retinal organoids¹⁹.”

7. For Figure 6, images B-D and F should be referred to in the main text.

RESPONSE. We have added references for 6B-D; 6F is referred to on page 15 line 328.

8. On page 15 (line 341), Supplementary Figure 11A-J should be stated, rather than 10A-I and on line 344, Supplementary Figure 11K-N should be referred to, instead of 10J-M. Again, on page 19 (line 419) Supplementary Figure 11 should be noted, not 10.

RESPONSE. These issues have been corrected – thanks for pointing them out.

9. Finally, the presence of MEF2C in cone photoreceptors in retinal organoids at day 160 and day 122 in human fetal retina does suggest a possible role in cone development. As it appears to be present in adult rod photoreceptors in mice, have the authors examined any later stage organoids (>200 days) for the presence and localisation of MEF2C, or has it been examined in adult human retinal sections at all?

RESPONSE. We have added panels O-R, MEF2C immunohistochemistry in adult monkey retina to supplementary figure 11. This shows strong MEF2C signal in cone nuclei and in cells in the INL reminiscent of Müller glia. In addition, there is very faint signal in rod nuclei. Thus it appears that in adult primates, MEF2C is cone (and possibly MG) enriched. We have modified lines 380-392 to read: “MEF2C expression was also detected in d122 human fetal retina (although not earlier) within a single cell layer adjacent to NR2E3+ developing rod nuclei, corresponding to the location where cones are found (Supplementary Fig. 11J-N). Strong MEF2C expression was also detected in peripheral adult monkey retina in infrequent cells next to NR2E3 positive rod nuclei, also corresponding to cones (Supplementary figure 11O-R) as well as within a line of INL nuclei likely corresponding to Müller glia. In addition, faint expression could be detected in rod nuclei (Supplementary figure 11Q). Additionally, analysis of published adult human scRNAseq data confirmed enrichment of MEF2C expression in cones relative to rods (p=0.04489, one-sided T test). This protein localization profile and adult human photoreceptor expression, as well as the enrichment both in expressed genes and differentially

expressed genes in our dataset, suggests a possible role for *MEF2C* in human cone photoreceptor development or maturation.”

Reviewer 3.

1) The assertion that reintroduction of NRL into the organoids restores rod formation is overstated. It does certainly reduce S OPSIN expression in those cells, but RHODOPSIN expression is negligible (only one convincingly RHO+ cell is shown), and no other markers are assessed. The authors should qualify their conclusions and ideally provide further evidence of other early rod markers following the introduction of NRL. It is not unreasonable that forced expression of NRL only partially rescues the rod phenotype, but this should be made clear.

The staining for RHODOPSIN is nicely specific in 3m but is unconvincing in 3r-t and it is not possible to discern whether there is or is not colocalization with ARR3. Panel s appears to be at a lower magnification to r and t making the supposed colocalization difficult to assess. Better images and staining are required along with single confocal sections. In 3k, o I assume that NRL labelling is immunostaining (no note of a GFP reporter in the NRL lentiviral construct), in which case why is it not localized to the nucleus? Can the authors comment on why RHO is so low given the relatively robust clusters of NRL+ cells (in “o”, at least). To what degree is RHO normally expressed in WT cultures of the same stage?

RESPONSE. We apologize and certainly did not mean to overstate our findings, but we understand the reviewer’s point. We have reworded our discussion (lines 402-404) to read: “This apparent perturbation of photoreceptor cell fate determination was accompanied by a dramatic shift in the ratio of S- to ML-opsin+ photoreceptors (Fig. 2). Additionally, we confirmed that this phenotype was due to lack of NRL by reintroducing a functional *NRL* and demonstrating restriction of S-opsin expression and promotion of RHO expression in mutant photoreceptors (Fig. 3).”

The staining for RHO in 3m outlines an immature cell where RHO is expressed throughout the cell body. The staining for RHO in 3s (which is the same magnification as all the panels in that row since they are separate channels of the same image) mostly shows bright RHO in the outer segments with only faint immunostaining in the cell bodies. We have cropped and re-sized the image to better convey the point. In addition, we have included a WT image of RHO staining in new supplementary figure 4 as requested by reviewer 2. In 3k, NRL is not confined to the nucleus because the transgene is under the control of a strong constitutive promoter so much higher than endogenous levels are being made in this cluster of cells. We note (line 159) that RHO+ cells were rare.

*2) The authors note differences in the *Nrl*^{-/-} mouse retina and hPSC-derived organoids with respect to OLM integrity and rosette formation. They suggest that the hiPSC-derived retinas do not form rosette because they are not spatially constrained. They should also note the work by Grimm and colleagues on the role that RPE65 plays in the formation of whorls/OLM disturbances in the *Nrl*^{-/-}. This is of interest given the lack of significant amounts of RPE in the organoids*

RESPONSE. As also noted in our response to Reviewer 2, comment #4, based on the comments of both reviewers, and so as not to distract from the major points of our manuscript, we have removed our comparative comments about rosette formation.

3) P18 - *The idea that the true cones survive, and the cods degenerate in the NRL null is particularly interesting. Is there evidence of degeneration in the L75Pfs organoids (caspase staining?), as might be predicted from the Nrl^{-/-} mice and the clinical phenotype. Related, can the authors comment on whether the MG in L75Pfs are in a more activated state compared to WT MG? Do the authors know if the true cones correctly locate to the apical most side or do they remain muddled with the cod like cells?*

RESPONSE. It would be unlikely to see degeneration in 200 day old organoids, since we would expect, based on the human time course, for degeneration to take longer. As per the reviewer's comments, we would indeed be very interested to be able to spatially distinguish the true cones from the cod-like cells, but so far we have not been able to. In terms of possible MG activation, we performed a 1-sided T test of GFAP expression in d170 L75Pfs vs WT muller glia and observed no upregulation. Additionally, based on immunostaining, we did not see any CRALBP+/GFAP+ MG in either the WT or mutant organoids, and the small number of caspase positive cells in the ONL of the WT and mutant organoids were similar.

Minor

4) P6, 1147 – was ARR3 found in the NRL+ cells? Unclear from images.

RESPONSE. NRL and S opsin co-localization was not observed as we did not co-stain for these markers.

5) P8, 1178 – did the authors mean “incomplete?”

RESPONSE. We did mean complete, as the 3' 75 bases of M and L opsin are identical, and thus indistinguishable given our sequencing parameters.

6) P10, 1214 – sentence is a little ambiguous. Please clarify.

RESPONSE. Apologies for not being clear. We have reworded the sentence to read: “Although WT and L75Pfs organoids had significantly different relative numbers of cells expressing *OPN1SW* vs *OPN1MW*, on an individual cell basis the *OPN1SW* and *OPN1MW* expressing cells expressed comparable levels of *OPN1SW* and *OPN1MW*.”

7) P22, 1494 – reference to Supplementary Table “Y”?

RESPONSE. Thanks for pointing this error out. We have corrected the reference to Supplementary Table 2.

8) Fig 2 – what is the red signal in 2a? What accounts for the missing 20% in the L75Pfs graphs?

RESPONSE. Since the antibodies against S and ML opsin were both derived in rabbit (see reference in Materials and Methods, lines 514-517), we directly labeled the ML antibody with AF555 (red) and used a rabbit green secondary to detect all opsin labeling. Thus S opsin was detected with green secondary and is green in the images while ML opsin was detected with green secondary while fluorescing red and appears orange in the images. Since the antigen is concentrated in the outer segments and at a much lower concentration in the cell body, the tips of the cells fluoresce yellow while the rest of the cell is orange.

9) *Fig 4 – it would be helpful to have the cell types coloured the same in 4A and 4C*

RESPONSE. We have changed the populations colors to match in 4A and 4C.

10) *Fig 6 – please make the colour dots for the different states larger so it is easier to identify the different populations referred to in the text.*

RESPONSE. As suggested, we have enlarged the size of the dots to improve clarity.

11) *Figures, it would be helpful to state the days in culture (d100 or d170) on the images on the different EBs*

RESPONSE. There are no images of EBs in the manuscript. The ages or stages of the organoids depicted are stated in the text and in the figure legends for figures 1 (line 115 and line 766) and 2 (line 120 and 772) and in the text for figure 3 (lines 149-151).

12) *Methods for counting cell number – the selection of 6 ROIs per organoid and 3 organoid per condition is quite limited given the established variability between EBs even within the same well. It is also not clear if these are made in a blinded manner. Although the reported magnitude of change is very large, and the conclusions are not in doubt, it would be better practice to implement a more robust sampling procedure for assessing no. of immuno-positive cells e.g. for Fig.2.*

RESPONSE. We agree with the reviewer and in the future will implement a more robust sampling procedure.

13) *Supp table 6 – 1154 has been autocorrected to a date.*

RESPONSE. This error has been corrected.

We would once again like to thank the reviewers for their very careful and detailed reviews and for their many insightful suggestions. We feel the requested revisions have significantly improved the manuscript, and hope that you will agree.

REVIEWERS' COMMENTS:

Reviewer #1 (Remarks to the Author):

The authors have addressed all of this reviewers previous concerns. This is an outstanding manuscript that presents a large body of high quality data that contributes greatly to the field.

Reviewer #2 (Remarks to the Author):

In my opinion, the authors have satisfactorily addressed all the points raised in the review process and I highly recommend the revised manuscript for publication in Communications Biology.

Reviewer #3 (Remarks to the Author):

The authors have addressed my comments adequately.